# Towards Generalizing Neural Topical Representations

## Abstract

Topic models have evolved from conventional Bayesian probabilistic models to recent Neural Topic Models (NTMs). Although NTMs have shown promising performance when trained and tested on a specific corpus, their generalization ability across corpora has yet to be studied. In practice, we often expect that an NTM trained on a source corpus can still produce quality topical representation (i.e., latent distribution over topics) for the document from different target corpora to a certain degree. In this work, we aim to improve NTMs further so that their representation power for documents generalizes reliably across corpora and tasks. To do so, we propose to enhance NTMs by narrowing the semantic distance between similar documents, with the underlying assumption that documents from different corpora may share similar semantics. Specifically, we obtain a similar document for each training document by text data augmentation. Then, we optimize NTMs further by minimizing the semantic distance between each pair, measured by the Topical Optimal Transport (TopicalOT) distance, which computes the optimal transport distance between their topical representations. Our framework can be readily applied to most NTMs as a plug-and-play module. Extensive experiments show that our framework significantly improves the generalization ability regarding neural topical representation across corpora.

## 1 Introduction

Topic modeling is a powerful technique for discovering semantic structures of text corpora in an unsupervised manner. It brings success to various applications, such as information retrieval (Blei & Jordan, 2003), marketing analysis (Reisenbichler & Reutterer, 2019), social media analysis (Laureate et al., 2023) and bioinformatics (Liu et al., 2016). Conventional topic models such as Latent Dirichlet allocation (LDA) (Blei et al., 2003) are Bayesian probabilistic models that assume generative stories of the data. With the increasing scale of data and the development of modern deep learning techniques, the union of deep learning and topic modeling, namely the Neural Topic Model (NTM) (Zhao et al., 2020), is becoming a popular technique for text analytics.

Given a collection of documents, a topic model learns a set of latent topics, each describing an interpretable semantic concept. A topic model is usually used in two ways: Using the topics to interpret the content of a corpus and using the topic distribution of a document as the semantic representation (i.e., topical representation). For the latter, the learned topical representations by topic models have shown good performance in downstream applications such as document classification (Nguyen & Luu, 2021), clustering (Zhao et al., 2020), and retrieval (Larochelle & Lauly, 2012). In practice, it is important that a trained model yields good representations for new documents. Ideally, these new documents are i.i.d. samples from the same distribution of the training documents (e.g., from the same corpus). However, this assumption is usually too strong for real-world applications, where new documents may not share the same data distribution with the training data (e.g., from different corpora). In this work, given an NTM trained on a source corpus, we are interested in how well its power of learning neural topical representation of documents generalizes to an unseen corpus without retraining. More importantly, we aim to propose a model-agnostic training scheme that can improve the generalization power of an arbitrary NTM. Although many methods have been proposed for generalizing deep neural networks to unseen domains (Wang et al., 2022; Zhou et al., 2022), most of them are designed for image data and cannot be applied to topic models. This is potentially because that topic models are

unsupervised methods whose latent representations (i.e., topics) encode specific semantic meanings and the evaluation of a model's generalization power is quite different from that of computer vision. Therefore, we believe that the problem studied in this work has not been carefully investigated in the literature.

Our idea is straightforward: If an NTM generalizes, it shall yield similar topical representations for documents with similar content. Based on this assumption, we further enhance NTMs by minimizing the distance between similar documents, which are created by text data augmentation (Wei & Zou, 2019; Shorten et al., 2021; Feng et al., 2021; Bayer et al., 2022). Specifically, a document can be encoded as a latent distribution $z$ over topics by NTMs. To make the model capable of producing quality $z$ for unseen documents, we encourage the model to learn similar $z$ for similar documents, which can be generated by document augmentations (Shorten et al., 2021; Bayer et al., 2022) such as adding, dropping and replacing words or sentences in the documents. To bring the topical representations of similar documents close together, we need to measure the distance between topical representations. This is done by a topical optimal transport distance that computes the distance between documents' topic distribution $z$. It naturally incorporates semantic information from topics and words into the distance computation between documents. Finally, with the optimal transport distance between the document and its augmentation, we propose to minimize this distance as a regularization term for training NTMs for better generalization. Our generalization regularization (Greg) term can be easily plugged into the training procedure of most NTMs. Our main contributions are summarized as followings:

- We are the first study of improving NTMs' generalization capability regarding document representation, which is expected in practice, especially for downstream tasks based on document representation.

- We introduce a universal regularization term for NTMs by applying text data augmentation and optimal transport, which brings consistent improvements over the generalization ability of most NTMs.

- We examine the generalization capability of NTMs trained on a source corpus by testing their topical representations on a different target corpus, which is a new setup for topic model evaluation.

## 2 Background

This section introduces the background of neural topic models and optimal transport, along with the notations used in this paper, which will facilitate the understanding of our method discussed in section 3. A summary of common math notations used in this paper is provided in Appendix A.

### 2.1 Neural Topic Models

Given a document collection, a topic model aims to learn the latent topics and the topical representation of documents. Specifically, a document in a text corpus $\mathcal{D}$ can be represented as a Bag-Of-Words (BOW) vector $x \in \mathbb{N}^V$, where $V$ denotes the vocabulary size (i.e., the number of unique words in the corpus). A topic model learns a set of $K$ topics $T := \{t_1, ..., t_K\}$ of the corpus, each $t_k \in \Delta^V$ is a distribution over the $V$ vocabulary words. The model also learns a distribution over the $K$ topics $z \in \Delta^K$ for document $x$ by modeling $p(z|x)$, where $z$ can be viewed as the topical representation of document $x$. To train a topic model, one usually needs to "reconstruct" the BOW vector by modeling $p(x|z)$.

Most conventional topic models such as LDA (Blei et al., 2003) are Bayesian probabilistic models, where $p(x|z)$ is built with probabilistic graphical models and inferred by a dedicated inference process. Alternatively, Neural Topic Models (NTMs) (Zhao et al., 2021) have been recently proposed, which use deep neural networks to model $p(z|x)$ and $p(x|z)$. Although there have been various frameworks for NTMs, models based on Variational Auto-Encoders (VAEs) (Kingma & Welling, 2013) and Amortized Variational Inference (AVI) (Rezende et al., 2014) are the most popular ones.

For VAE-NTMs, $p_\phi(x|z)$ is modeled by a decoder network $\phi$: $x' := \phi(z)$; $p(z|x)$ is approximated by the variational distribution $q_\theta(z|x)$ which is modeled by an encoder network $\theta$: $z := \theta(x)$. The learning objective

of VAE-NTMs is maximizing the Evidence Lower Bound (ELBO):

$$\max_{\theta,\phi} \left( \mathbb{E}_{q_\theta(\boldsymbol{z}|\boldsymbol{x})}[\log p_\phi(\boldsymbol{x}|\boldsymbol{z})] - \mathbb{KL}[q_\theta(\boldsymbol{z}|\boldsymbol{x}) \parallel p(\boldsymbol{z})] \right), \tag{1}$$

where the first term is the conditional log-likelihood, and the second is the Kullback–Leibler (KL) divergence between the variational distribution of $\boldsymbol{z}$ and its prior distribution $p(\boldsymbol{z})$. By using one linear layer for the decoder in NTMs, one can obtain the topic over word distributions by normalizing the columns of the decoder's weight $W \in \mathbb{R}^{V \times K}$. Note that although VAE-NTMs are of the most interest in this paper, our proposed method is not specifically designed for them, it can be applied to other NTM frameworks as well.

## 2.2 Optimal Transport

Optimal Transport (OT) has been widely used in machine learning for its capability of comparing probability distributions (Peyré et al., 2019). Here, we focus on the case of OT between discrete distributions. Let $\mu(\boldsymbol{X},\boldsymbol{a}) := \sum_{i=1}^{N} \boldsymbol{a}_i \delta_{\boldsymbol{x}_i}$ and $\mu(\boldsymbol{Y},\boldsymbol{b}) := \sum_{j=1}^{M} \boldsymbol{b}_j \delta_{\boldsymbol{y}_j}$, where $\boldsymbol{X} := \{\boldsymbol{x}_1, \cdots, \boldsymbol{x}_N\}$ and $\boldsymbol{Y} := \{\boldsymbol{y}_1, \cdots, \boldsymbol{y}_M\}$ denote the supports of the two distributions, respectively; $\boldsymbol{a} \in \Delta^N$ and $\boldsymbol{b} \in \Delta^M$ are probability vectors in $\Delta^N$ and $\Delta^M$, respectively. The OT distance between $\mu(\boldsymbol{X},\boldsymbol{a})$ and $\mu(\boldsymbol{Y},\boldsymbol{b})$ can be defined as:

$$D_{\boldsymbol{M}} \left( \mu(\boldsymbol{X},\boldsymbol{a}), \mu(\boldsymbol{Y},\boldsymbol{b}) \right) := \inf_{\boldsymbol{P} \in U(\boldsymbol{a},\boldsymbol{b})} \langle \boldsymbol{P}, \boldsymbol{M} \rangle, \tag{2}$$

where $\langle \cdot, \cdot \rangle$ denotes the Frobenius dot-product; $\boldsymbol{M} \in \mathbb{R}_{\geq 0}^{N \times M}$ is the cost matrix of the transport which defines the pairwise cost between the supports; $\boldsymbol{P} \in \mathbb{R}_{>0}^{N \times M}$ is the transport matrix; $U(\boldsymbol{a},\boldsymbol{b})$ denotes the transport polytope of $\boldsymbol{a}$ and $\boldsymbol{b}$, which is the polyhedral set of $N \times M$ matrices:

$$\boldsymbol{U}(\boldsymbol{a},\boldsymbol{b}) := \left\{ \boldsymbol{P} \in \mathbb{R}_{>0}^{N \times M} | \boldsymbol{P}\mathbf{1}_M = \boldsymbol{a}, \boldsymbol{P}^T\mathbf{1}_N = \boldsymbol{b} \right\}, \tag{3}$$

where $\mathbf{1}_M$ and $\mathbf{1}_N$ are the $M$ and $N$-dimensional column vector of ones, respectively. The OT distance can be calculated by finding the optimal transport plan $\boldsymbol{P}^*$, for which various OT solvers (Flamary et al., 2021) have been proposed. The direct optimization of Eq. (2) is computationally expensive, Cuturi (2013) introduced the entropy-regularized OT distance, known as the Sinkhorn distance, which is more efficient for large-scale problems. It can be defined as:

$$D_{\boldsymbol{M},\lambda}(\mu(\boldsymbol{X},\boldsymbol{a}), \mu(\boldsymbol{Y},\boldsymbol{b})) := \inf_{\boldsymbol{P} \in U_\lambda(\boldsymbol{a},\boldsymbol{b})} \langle \boldsymbol{P}, \boldsymbol{M} \rangle, \tag{4}$$

where $\boldsymbol{U}_\lambda(\boldsymbol{a},\boldsymbol{b})$ defines the transport plan with the constraint that: $h(\boldsymbol{P}) \geq h(\boldsymbol{a}) + h(\boldsymbol{b}) - \lambda$, where $h(\cdot)$ denotes the entropy function and $\lambda \in [0,\infty]$ is the hyperparameter.

## 3 Method

### 3.1 Problem Setting

In this paper, given an NTM trained on a source corpus $\mathcal{D}^S$, we are interested in how to train a neural topic model on $\mathcal{D}^S$ so that it can generate good topical representations not only on $\mathcal{D}^S$ but also on unseen corpora *without* retraining on them. We note that an NTM with a standard training scheme usually has some intrinsic generalization power to new corpora, as documents from different domains may share similar semantics. However, we argue that such intrinsic generalization power is not enough to generate quality topical representations for unseen documents. In this work, we aim to study and improve such intrinsic power of NTMs.

### 3.2 Overview of the Proposed Method

We enhance NTMs' generalization by assuming that a document's topical representation should be close to the topical representations of its augmentations. Specifically, let $\boldsymbol{x}^s$ be the BOW vector of the document from the source corpus. Suppose a stochastic function $\mathcal{F}$ can produce a random augmentation $\boldsymbol{x}^{aug}$ that

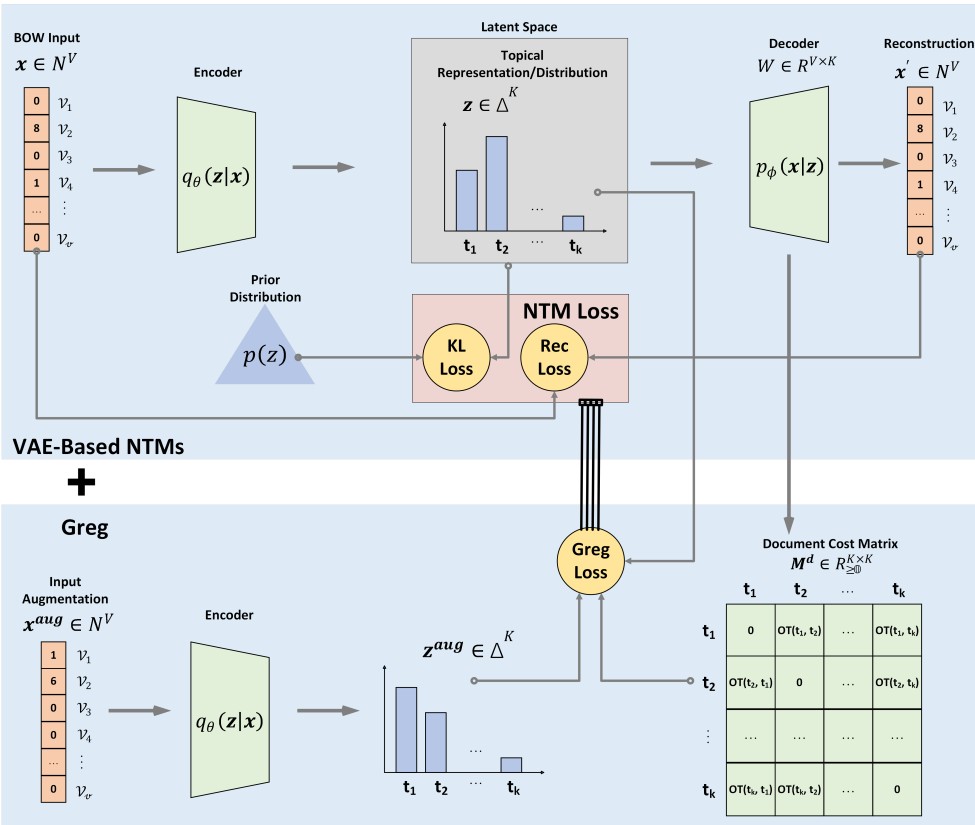

Figure 1: Neural Topic Model (NTM) with Generalization Regularization (Greg). The BOW vectors of a document and its augmentation are encoded as the topical representations, respectively; Besides common VAE-NTMs that aim to reconstruct the input ("Rec Loss") and match the posterior distribution to the prior ("KL Loss"), we encourage the model to produce a similar $\boldsymbol{z}$ for the original document and its augmentation; The distance between $\boldsymbol{z}$ is measured by TopicalOT as our "Greg Loss", which is guided by the document cost matrix whose entries specify the OT cost of moving between topics. Our framework can be readily applied to most NTMs as a plug-and-play module. (We draw two encoders here for tidy illustration; they are identical.)

is semantically similar to $\boldsymbol{x}^s$: $\boldsymbol{x}^{aug} := \mathcal{F}(\boldsymbol{x}^s)$. As both $\boldsymbol{x}^s$ and $\boldsymbol{x}^{aug}$ share similar semantics, their topical representations $\boldsymbol{z}^s$ and $\boldsymbol{z}^{aug}$ should stay close in the learned representation space. To achieve this, we introduce a new semantics-driven regularization term to the existing training objective of an NTM, which additionally minimizes the distance between $\boldsymbol{z}^s$ and $\boldsymbol{z}^{aug}$ with respect to the encoder parameters $\theta$:

$$\min_{\theta} D(\boldsymbol{z}^s, \boldsymbol{z}^{aug}). \tag{5}$$

Straightforward choices of $D$ can be the Euclidean, Cosine, or Hellinger distances, to name a few. However, these distances cannot sufficiently capture the semantic distance between documents in terms of their topic distributions. To address this issue, we propose to use a topical optimal transport distance between documents, inspired by the Hierarchical OT (HOT) distance from Yurochkin et al. (2019). We refer to it as TopicalOT throughout our paper for clarity within our context. Specifically, we are given the word embedding matrix $\boldsymbol{E} \in \mathbb{R}^{V \times L}$ from pre-trained models such as Word2Vec (Mikolov et al., 2013), GloVe (Pennington et al., 2014), BERT (Devlin et al., 2018) and etc., where $V$ is the vocabulary size and $L$ is the embedding dimension. Each word embedding is denoted by $\boldsymbol{e}^v \in \mathbb{R}^L$, where $v \in [1, \ldots, V]$ denotes the $v$-th word in the vocabulary. Let $\boldsymbol{T} := \{\boldsymbol{t}_1, ..., \boldsymbol{t}_K\} \subset \Delta^V$ be the learned topics of the corpus, each of which is a distribution over words. Therefore, each topic $\boldsymbol{t}_k$ can be viewed as a discrete distribution whose supports

are the word embedding: $\mu(\boldsymbol{E}, \boldsymbol{t}_k)$. Then, the OT distance between topic $\boldsymbol{t}_{k_1}$ and $\boldsymbol{t}_{k_2}$ can be computed by:

$$
\begin{aligned}
\mathrm{WMD}(\boldsymbol{t}_{k_1}, \boldsymbol{t}_{k_2}) &:= D_{\boldsymbol{M}^t}(\mu(\boldsymbol{E}, \boldsymbol{t}_{k_1}), \mu(\boldsymbol{E}, \boldsymbol{t}_{k_2})), \\
\text{where } \boldsymbol{M}^t_{v_1, v_2} &:= 1 - \cos(\boldsymbol{e}^{v_1}, \boldsymbol{e}^{v_2}),
\end{aligned}
\tag{6}
$$

where $D_{\boldsymbol{M}^t}(\cdot, \cdot)$ denotes the OT distance in Eq. (2); $\boldsymbol{M}^t \in \mathbb{R}^{V \times V}_{\geq 0}$ is the topic cost matrix and $\cos(\cdot, \cdot)$ denotes the Cosine similarity; $k_1, k_2 \in [1, ..., K]$ are indices of topics and $v_1, v_2 \in [1, ..., V]$ are indices of vocabulary words. Eq. (6) can be regarded as the Word Mover's Distance (WMD) (Kusner et al., 2015), measuring the topic distance instead of the document distance.

Similarly, as the document's topical representation $\boldsymbol{z}$ can be viewed as a discrete distribution whose supports are the topics, the OT Distance between documents $\boldsymbol{d}_i$ and $\boldsymbol{d}_j$ is computed by:

$$
\begin{aligned}
\mathrm{TopicalOT}(\boldsymbol{d}_i, \boldsymbol{d}_j) &:= D_{\boldsymbol{M}^d}(\mu(\boldsymbol{T}, \boldsymbol{z}_i), \mu(\boldsymbol{T}, \boldsymbol{z}_j)) \\
\text{where } \boldsymbol{M}^d_{k_1, k_2} &:= \mathrm{WMD}(\boldsymbol{t}_{k_1}, \boldsymbol{t}_{k_2}),
\end{aligned}
\tag{7}
$$

where $\boldsymbol{M}^d \in \mathbb{R}^{K \times K}_{\geq 0}$ is the document cost matrix whose entries indicate the OT distance between topics as in Eq. (6).

As for topics, we use the decoder weights: $W \in \mathbb{R}^{V \times K}$ as the representation of topics, like other NTMs. Since OT measures the distance between distributions, we normalize each topic as the topic over word distribution by the softmax function:

$$
\boldsymbol{t}_k := \mathrm{softmax}(W^{\mathrm{T}}_{k,:}),
\tag{8}
$$

where T denotes the matrix transpose operation. Then, we can construct $\boldsymbol{M}^d$ by computing the OT distance between topics as Eq. (6). Putting all together, we have:

$$
\min_{\theta, W} D_{\boldsymbol{M}^d}(\mu(\boldsymbol{T}, \boldsymbol{z}^s), \mu(\boldsymbol{T}, \boldsymbol{z}^{aug})),
\tag{9a}
$$

$$
\text{where } \boldsymbol{T} := \{\boldsymbol{t}_1, ..., \boldsymbol{t}_K\}, \ \boldsymbol{z} := \theta(\boldsymbol{x}),
$$

$$
\boldsymbol{M}^d_{k_1, k_2} := D_{\boldsymbol{M}^t}(\mu(\boldsymbol{E}, \boldsymbol{t}_{k_1}), \mu(\boldsymbol{E}, \boldsymbol{t}_{k_2})),
\tag{9b}
$$

$$
\boldsymbol{M}^t_{v_1, v_2} := 1 - \cos(\boldsymbol{e}^{v_1}, \boldsymbol{e}^{v_2}).
\tag{9c}
$$

Intuitively, we are encouraging the model to produce similar topical representations for documents with similar semantics, where the semantic similarity between topical representations is captured by a two-level OT distance. At the document level, TopicalOT compute the OT distance between two topical representations of documents, where the cost matrix is defined by the semantic distances between topics. At the topic level, the distance between two topics is again measured by OT, where the transport cost is determined by distances between word embeddings. The whole computation injects rich semantic information from both external (i.e., word embeddings from the pre-trained model) and internal (i.e., the topic model itself) sources, thus better capturing the semantic similarity between documents.

### 3.3 Efficiently Computing Topical OT

Directly solving the problem in Eq. (9) during training is computationally expensive for two reasons: (i) To obtain $\boldsymbol{M}^d$ in Eq. (9b), we need to compute the OT distance between $K \times (K - 1)/2$ topic pairs, each of them has a problem space of $V \times V$. This will be expensive since a corpus may have a large vocabulary size; (ii) NTMs are usually trained on a large text corpus in batches of documents $\boldsymbol{X} := \{\boldsymbol{x}_i\}^B_{i=1} \in \mathbb{N}^{B \times V}$ where $B$ denotes the batch size, so we need to compute between $\boldsymbol{Z}^s$ and $\boldsymbol{Z}^{aug}$ where $\boldsymbol{Z} := \{\boldsymbol{z}_i\}^B_{i=1} \in \mathbb{R}^{B \times K}$. While the original HOT leverages the algorithm in Bonneel et al. (2011) for the computation of OT, which can not support the computation between $\boldsymbol{z}$ pairs of $\boldsymbol{Z}^s$ and $\boldsymbol{Z}^{aug}$ in parallel, thus causing an enormous computational cost during training.

To address the first issue, we leverage the fact that a topic's semantics can be captured by its most important words. Specifically, when computing $\boldsymbol{M}^d$, we reduce the dimension of each topic $\boldsymbol{t}_k$ by considering only the

top $I$ words that have the largest weights in the topic:

$$\tilde{\boldsymbol{t}}_k := f_N(f_I((\boldsymbol{t}_k)), \tag{10}$$

where $\boldsymbol{t}_k$ is the topic before approximation defined in Eq. (8); $f_I$ is a function that returns the subset that contains $I$ elements with the largest weights; $f_N$ denotes the function for re-normalizing by dividing by the sum. Now, we reduce the problem space of solving OT between one topic pair in Eq. (9b) from $V \times V$ to $I \times I$, and each estimated topic $\tilde{\boldsymbol{t}}_k$'s related vocabulary and word embeddings become $\mathcal{V}^{\tilde{\boldsymbol{t}}_k}$ and $\boldsymbol{E}^{\tilde{\boldsymbol{t}}_k}$, respectively. Since the vocabulary of each topic become different then, the topic cost matrix in Eq. (9c) will vary for different topic pairs, which is denoted by $\boldsymbol{M}^{\tilde{\boldsymbol{t}}_{k_1}, \tilde{\boldsymbol{t}}_{k_2}} \in \mathbb{R}_{\geq 0}^{I \times I}$ for topic $\tilde{\boldsymbol{t}}_{k_1}$ and $\tilde{\boldsymbol{t}}_{k_2}$. Now, we rewrite Eq. (9b) and Eq. (9c) as:

$$\boldsymbol{M}_{k_1,k_2}^{\boldsymbol{d}} := D_{\boldsymbol{M}^{\tilde{t}_{k_1}, \tilde{t}_{k_2}}}(\mu(\boldsymbol{E}^{\tilde{\boldsymbol{t}}_{k_1}}, \tilde{\boldsymbol{t}}_{k_1}), \mu(\boldsymbol{E}^{\tilde{\boldsymbol{t}}_{k_2}}, \tilde{\boldsymbol{t}}_{k_2})), \tag{11a}$$

$$\text{where } \boldsymbol{M}_{v_1,v_2}^{\tilde{\boldsymbol{t}}_{k_1}, \tilde{\boldsymbol{t}}_{k_2}} := 1 - \cos(\boldsymbol{e}^{v_1}, \boldsymbol{e}^{v_2}). \tag{11b}$$

So far, we reduce the size of topic cost matrix to $I \times I$ (i.e., $v_1, v_2 \in [1, ..., I]$). We approximate this way because only a small subset of most important words is helpful for the understanding of a topic, similar to the consideration when evaluating topic coherence (Newman et al., 2010; Lau et al., 2014).

To address the second issue, we replace the OT distance between $\boldsymbol{z}$ with the Sinkhorn distance defined in Eq. (4), and leverage the Sinkhorn algorithm (Cuturi, 2013) for its efficiency and parallelization. As for the distance between topics, we keep it as OT distance because each topic does not share the same vocabulary set by our approximation approach, which results in different cost matrices for each topic pair. Although it has to be computed pairwise for topic pairs, it is still feasible since the number of topics $K$ is usually small in the settings of NTMs. Putting all together, we have the following as the final form for efficiently computing TopicalOT during training:

$$\min_{\theta, W} D_{\boldsymbol{M}^d, \lambda}(\mu(\tilde{\boldsymbol{T}}, \boldsymbol{Z}^s), \mu(\tilde{\boldsymbol{T}}, \boldsymbol{Z}^{aug})), \tag{12a}$$

$$\text{where } \tilde{\boldsymbol{T}} := \{\tilde{\boldsymbol{t}}_1, ..., \tilde{\boldsymbol{t}}_K\}, \ \boldsymbol{Z} := \theta(\boldsymbol{X}),$$

$$\boldsymbol{M}_{k_1,k_2}^{\boldsymbol{d}} := D_{\boldsymbol{M}^{\tilde{t}_{k_1}, \tilde{t}_{k_2}}}(\mu(\boldsymbol{E}^{\tilde{\boldsymbol{t}}_{k_1}}, \tilde{\boldsymbol{t}}_{k_1}), \mu(\boldsymbol{E}^{\tilde{\boldsymbol{t}}_{k_2}}, \tilde{\boldsymbol{t}}_{k_2})), \tag{12b}$$

$$\boldsymbol{M}_{v_1,v_2}^{\tilde{\boldsymbol{t}}_{k_1}, \tilde{\boldsymbol{t}}_{k_2}} := 1 - \cos(\boldsymbol{e}^{v_1}, \boldsymbol{e}^{v_2}). \tag{12c}$$

Compared to the original HOT distance, which only serves as a distance measure between documents, our approximation by TopicalOT supports the computation of distances between batches of document pairs in parallel. Additionally, it is optimizable, allowing for easy integration during model training.

### 3.4 Document Augmentation (DA)

As for the function $\mathcal{F}(\cdot)$ that generates a random augmentation $\boldsymbol{x}^{aug}$ of the original document $\boldsymbol{x}^s$, our framework is agnostic to the text augmentation approach employed and is not limited to those discussed in Wei & Zou (2019); Shorten et al. (2021); Feng et al. (2021); Bayer et al. (2022). As summarized in Bayer et al. (2022), document augmentation can occur at the character, word, phrase, or document level. Since common NTMs are trained on BOWs, we focus on word-level augmentation, which can be efficiently integrated during training. Different word-level (Ma, 2019) document augmentations are investigated and their descriptions are summarized in Table 1. Their effect to our generalization framework are studied in Appendix D.3. As a general form, here we write $\boldsymbol{z}^{aug}$ is obtained by:

$$\boldsymbol{z}^{aug} := \text{softmax}(\theta(\mathcal{F}(\boldsymbol{x}^s, \beta, \Omega))), \tag{13}$$

where $\beta$ is the augmentation strength that determines the number of words to be varied: $n = \text{ceil}(\beta \times l)$ where $l$ is the document length and $\text{ceil}(\cdot)$ rounds up the given number; $\Omega$ denotes other information needed for the augmentation, such as the number of top words for replacement, and the pre-trained word embeddings $\boldsymbol{E}$ to provide similarity between words.

Table 1: Word-level document augmentations

| Approach | Description |
|---|---|
| Random Drop | Randomly sample $n$ words from the document and drop; |
| Random Insertion | Randomly sample $n$ words from the vocabulary and add to the document; |
| Random to Similar | Randomly replace $n$ words of the document with one of their top similar words within the vocabulary; |
| Highest to Similar | Replace $n$ words with the highest Term Frequency–Inverse Document Frequency (TF-IDF) weight with one of their top similar words within the vocabulary; |
| Lowest to Similar | Replace $n$ words with the lowest TF-IDF weight with one of their top similar words within the vocabulary; |
| Random to Dissimilar | Randomly replace $n$ words of the document with one of their top dissimilar words within the vocabulary; |
| Highest to Dissimilar | Replace $n$ words with the highest TF-IDF weight with one of their top dissimilar words within the vocabulary; |
| Lowest to Dissimilar | Replace $n$ words with the lowest TF-IDF weight with one of their top dissimilar words within the vocabulary; |

### 3.5 Integrating Greg to the Training of Existing NTMs

The integration of Greg with existing NTMs is illustrated in Figure 1, the algorithm of Greg is provided in Appendix B. With the primary goal of NTMs that aims to reconstruct the input documents and match the posterior to the prior distribution, we propose the following joint loss:

$$\min_{\theta,\phi}(\gamma \cdot \mathbb{E}_{q_\theta(\boldsymbol{z}^s|\boldsymbol{x})}[D_{\boldsymbol{M^d},\lambda}(\mu(\tilde{\boldsymbol{T}}, \boldsymbol{z}^s), \mu(\tilde{\boldsymbol{T}}, \boldsymbol{z}^{aug}))] + \mathcal{L}^{\text{NTM}}), \tag{14}$$

where $\mathcal{L}^{\text{NTM}}$ is the original training loss of an NTM, which can be ELBO in Eq. (1) or other losses; The first term is the proposed Greg loss, where $\boldsymbol{z}^s := \text{softmax}(\theta(\boldsymbol{x}^s))$; $\boldsymbol{z}^{aug}$ is obtained by Eq. (13); $\boldsymbol{M^d}$ is parameterized by $\phi$[1] and can be obtained by solving Eq. (12b) and Eq. (12c); $\gamma$ is the hyperparameter that determines the strength of the regularization; $\lambda$ is the hyperparameter for the Sinkhorn distance. The training algorithm of our generalization regularization (Greg) is summarized in Algorithm 1. Notably, both the Sinkhorn distance and OT distance support auto differentiation in deep learning frameworks such as PyTorch (Patrini et al., 2020; Bonneel et al., 2011), thus the loss in Eq. (12) is differentiable in terms of $\theta$ and $\phi$.

## 4 Related Work

### 4.1 Neural Topic Models

For a comprehensive review of NTMs, we refer the readers to Zhao et al. (2021). Here, we mainly focus on models based on VAE (Kingma & Welling, 2013) and AVI (Rezende et al., 2014). Early works of NTMs focus on studying the prior distributions for the latent variables, such as Gaussian (Miao et al., 2017) and various approximations of the Dirichlet prior (Srivastava & Sutton, 2017; Zhang et al., 2018; Burkhardt & Kramer, 2019) for its difficulty in reparameterization (Tian et al., 2020). Recent NTMs mainly leverage external information such as complementary metadata (Card et al., 2017) and contextual embeddings (Dieng et al., 2020; Bianchi et al., 2020a;b; Xu et al., 2022). In this work, we are interested in generalizing NTMs instead of proposing a new NTM. We believe that our method is general to improve the generalization of most NTMs not limited to VAE-NTMs.

### 4.2 Topic Models and Optimal Transport

Recently, a few works have built the connection between topic modeling and OT, most focusing on developing new OT frameworks for topic modeling, such as non-neural topic model (Huynh et al., 2020) and NTMs

---

[1]Precisely, $\boldsymbol{M^d}$ is parameterized by $W$ which is the weight of the linear layer of $\phi$.

(Zhao et al., 2020; Nan et al., 2019; Zhang & Lauw, 2023). Our method is not an OT framework for NTMs but a general regularization term to improve the generalization of NTMs, which is also compatible with NTMs based on the OT frameworks.

### 4.3 Topic Model Generalization

Model generalization is a popular topic in machine learning. However, the generalization of topic models, especially NTMs, has not been comprehensively studied. (i) Most existing works focus on generalizing topic models to multiple domains with access to the full or partial data of the new domains for retraining/fine-tuning, such as the models on continual lifelong learning (Chen & Liu, 2014; Chen, 2015; Blum & Haghtalab, 2016; Chen et al., 2019; Gupta et al., 2020; Qin et al., 2021; Zhang et al., 2022; Lei et al., 2023) and few-shot learning (Iwata, 2021; Duan et al., 2022; Xu et al., 2024). While our approach needs no access to the data in the new domains nor retraining of the model. (ii) Some approaches focus on the generalization of topics across different languages under the zero-shot or few-short setting (Bianchi et al., 2020b; Chang & Hwang, 2021; Grootendorst, 2022). While ours focuses on the generalization of topical representation of unseen documents. (iii) Recent Large Language Model (LLM) based topic models (Wang et al., 2023; Pham et al., 2023; Chang et al., 2024) extract or refine topics by prompting, which inherits the generalization power of LLMs. Their generalization capability originates from numerous training data, supervised fine-tuning, etc., of LLMs, which is different from our setting as we only train the NTM on the source domain. Moreover, they primarily concentrate on topic extraction and allocate less emphasis to representing the documents themselves. (iv) In terms of domain generalization (Wang et al., 2022; Zhou et al., 2022), extensive works exist for computer vision tasks, including image classification (Yue et al., 2019; Liu et al., 2021), semantic segmentation (Gong et al., 2019; Li et al., 2021) and action recognition (Li et al., 2017; 2019); as well as some natural language processing tasks, such as sentiment classification (Balaji et al., 2018; Wang et al., 2020b) and semantic parsing (Wang et al., 2020a). To the best of our knowledge, there is no universal module or regularizer known to be applicable to topic models for domain generalization, probably due to topic models' unsupervised nature and the lack of comprehensive evaluation. Ours is the first specialized for NTMs' generalization. (v) Regarding the learning strategy, our work is related to the contrastive learning framework (Chen et al., 2020a), but we focus on only the positive pairs and leverage semantic distance between documents. As for the work that most related to ours, the Contrastive Neural Topic Model (CLNTM) proposed by Nguyen & Luu (2021) employs contrastive distance (Chen et al., 2020b) to regularize the topical representation of documents. However, there are key differences between CLNTM and our model. First, CLNTM does not emphasize the generalization of NTMs. Second, it measures the distance between document representations using Cosine distance, whereas our model utilizes TopicalOT, which integrates semantic information from both topics and words. Beyond the distinctions noted above, we anticipate that our approach will serve as a general regularization method to enhance other NTMs.

## 5 Experiments

### 5.1 Experimental Settings

#### 5.1.1 Datasets

We conduct our experiments on five widely-used datasets: 20 Newsgroup (**20News**) (Lang, 1995), **R8**[2], Web Snippets (**Webs**) (Phan et al., 2008), Tag My News (**TMN**) (Vitale et al., 2012) and **DBpedia** (Zhang et al., 2015). We pre-process the documents as BOW vectors by the following steps: We clean the documents by removing special characters and stop words, followed by tokenization. Then we build the vocabulary by considering the words with document frequency greater than five and less than 80% of the total documents. As we use word embeddings of GloVe (Pennington et al., 2014) pre-trained on Wikipedia[3], we filter the vocabulary words by keeping only the words that appear in the vocabulary set of GloVe. Finally, we convert documents to BOW vectors based on the final vocabulary set. The statistics of the pre-processed datasets

---

[2]https://www.kaggle.com/datasets/weipengfei/ohr8r52
[3]https://nlp.stanford.edu/projects/glove/

are summarized in Table C1. These datasets are further randomly split as training and testing sets by 8:2 for our experiments.

### 5.1.2 Evaluation Protocol

**Topical Representation Quality** We focus on the evaluation[4] of the quality of documents' topical representations, which is done by downstream tasks where the topical representations are used as input features for document classification and clustering: (i) Document Classification: We use the trained topic model to infer the topic distributions of the training and testing documents as their topical representations. Then we train a random forest classifier using the training documents' topical representation and evaluate the Classification Accuracy (**CA**) on testing documents. The random forest classifier consists of ten decision trees with a maximum depth of 8, which has the same setting as the previous work in Nguyen & Luu (2021). (ii) Document Clustering: We evaluate the clustering performance of test documents' topical representation based on the widely-used metrics, Purity and Normalized Mutual Information (NMI). Following Nguyen et al. (2015), we assign each test document to the cluster corresponding to its highest-weighted topic. Then, we compute Purity and NMI (denoted by Top-Purity (**TP**) and Top-NMI (**TN**), respectively) based on the cluster assignments of the documents and their true labels.

**Topical Representation Generalization** To evaluate the generalization of topic representations, we train a model on the source corpus and test on a target corpus. We explore two configurations of the target corpus: (i) from a different domain of the source corpus; (ii) from the same domain of the source corpus but with noise. When the targets are different text corpora, they may not share the same vocabulary sets, as different pre-processing may result in various subsets of the language's vocabulary for specific text corpus. This makes NTM generalization harder as an NTM can not accept input documents with a vocabulary set different from the training vocabulary. To address this issue, we unite the vocabulary set of all corpora for BOW representation during the training, allowing NTMs to accept input documents from different corpora (with varying vocabulary sets). When the target is the noisy source corpus, the noisy versions are created by randomly sampled compositions of document augmentations described in Table 1, where the augmentation strength is set as 0.75 (e.g., changing 75% of words in the original document). Notably, for all source-to-target tasks, the models are only trained on the source corpus with no access to the documents of the target corpus.

### 5.1.3 Backbones and Settings

Our generalization regularization (**Greg**) framework can be easily applied to most of the NTMs. Here, we consider the following popular NTMs as our backbones: (i) Neural Variational Document Model (**NVDM**) (Miao et al., 2017), a pioneer NTM that applies Gaussian prior to $z$. (ii) LDA with Products of Experts (**ProdLDA**) (Srivastava & Sutton, 2017), an NTM that replaces the mixture of multinomials in LDA with the product of experts. (iii) Neural Topic Model with Covariates, Supervision, and Sparsity (**SCHOLAR**) (Card et al., 2017), an NTM which applies logistic normal prior and incorporates metadata. (iv) Contrastive Neural Topic Model (**CLNTM**) (Nguyen & Luu, 2021), a recent NTM that uses a contrastive learning framework to regularize the document representation. We follow the default settings for all these models, except the learning rates are fine-tuned to fit our own datasets. As for Greg, different document augmentations can be used; we use "Highest to Similar" in Table 1 throughout our experiments where the top 20 similar words are considered for replacement. The justification for this choice is described in Section 5.3.1. As for hyperparameters of Greg, we set $\gamma$ as 300, $\beta$ as 0.5 for all experiments; As for the Sinkhorn algorithm (Cuturi, 2013), we fix $\lambda$ as 100, the maximum number of iterations as 5,000 and the stop threshold as 0.005; As for the OT distance, we leverage the function with default settings in the POT package[5] (Bonneel et al., 2011) for the calculation.

---

[4]All experiments in this paper are conducted five times with different random seeds. Mean and std values (in percentage) of metrics are reported.

[5]https://pythonot.github.io/index.html

Table 2: One source (20News) to different targets

| (%,↑) | Method | Target | | | |
|---|---|---|---|---|---|
| | | Webs | TMN | DBpedia | R8 |
| CA | NVDM | 39.0±0.8 | 38.8±0.5 | 31.2±0.6 | 73.2±0.6 |
| | + Greg | **63.7±0.3** | **60.2±0.4** | **55.8±0.4** | **80.6±0.5** |
| | PLDA | 25.7±0.5 | 32.0±0.6 | 14.9±0.8 | 59.8±0.7 |
| | + Greg | **31.6±0.4** | **40.0±0.6** | **19.0±0.4** | **61.2±1.1** |
| | SCHOLAR | 56.0±0.7 | 51.8±0.3 | 50.9±1.2 | **78.7±1.5** |
| | + Greg | **59.6±1.9** | **57.8±1.8** | **53.3±1.3** | 77.2±0.5 |
| | CLNTM | 46.5±1.7 | 43.8±1.1 | 42.0±1.4 | 74.9±1.9 |
| | + Greg | **54.5±2.2** | **54.0±1.6** | **47.8±0.9** | **76.7±1.3** |
| TP | NVDM | 28.6±0.7 | 32.2±0.4 | 19.7±0.7 | 62.1±2.1 |
| | + Greg | **35.8±1.5** | **40.1±1.5** | **26.3±0.9** | **63.7±1.8** |
| | PLDA | 24.7±0.5 | 26.9±0.2 | 13.0±0.3 | 56.4±0.3 |
| | + Greg | **25.7±0.8** | **29.5±0.8** | **14.8±0.4** | **57.7±0.8** |
| | SCHOLAR | 37.0±2.2 | 38.6±1.6 | 22.7±0.6 | **61.2±1.1** |
| | + Greg | **41.4±4.0** | **45.6±4.1** | **23.1±2.6** | 61.0±1.2 |
| | CLNTM | 31.8±2.8 | 37.0±1.9 | 20.8±1.5 | 63.6±2.6 |
| | + Greg | **34.3±1.7** | **41.9±2.5** | **21.1±1.9** | 63.6±4.7 |
| TN | NVDM | 6.3±0.3 | 3.8±0.1 | 9.0±0.5 | 13.3±1.1 |
| | + Greg | **12.7±0.8** | **9.6±0.8** | **15.7±0.5** | **15.5±0.7** |
| | PLDA | 3.6±0.2 | 1.6±0.1 | 4.0±0.2 | 7.7±0.3 |
| | + Greg | **4.3±0.3** | **2.7±0.4** | **5.1±0.3** | **8.4±0.2** |
| | SCHOLAR | 16.2±0.9 | 11.3±1.1 | 15.1±0.8 | **15.2±0.9** |
| | + Greg | **19.5±2.7** | **19.3±3.2** | **15.6±2.5** | 13.7±1.8 |
| | CLNTM | 10.1±1.7 | 7.9±1.0 | 11.3±0.9 | 15.3±2.4 |
| | + Greg | **14.1±1.8** | **14.6±2.6** | **12.4±1.5** | **15.4±4.5** |

Table 3: Different sources to one target (TMN)

| (%,↑) | Method | Target (TMN) | | | |
|---|---|---|---|---|---|
| | | 20News_T | Webs_T | DBpedia_T | R8_T |
| CA | NVDM | 38.4±0.3 | 44.4±0.3 | 39.5±1.8 | 33.5±0.6 |
| | + Greg | **59.6±0.4** | **63.4±0.6** | **59.4±0.6** | **43.3±0.7** |
| | PLDA | 31.7±0.1 | 43.1±0.6 | 37.4±0.6 | 26.8±0.2 |
| | + Greg | **40.7±0.5** | **48.8±0.8** | **42.0±0.3** | **27.3±0.1** |
| | SCHOLAR | 51.3±0.6 | 49.3±0.8 | 56.8±0.7 | 43.8±0.7 |
| | + Greg | **58.5±1.8** | **63.6±1.0** | **60.9±1.4** | **45.9±0.7** |
| | CLNTM | 44.8±1.3 | 45.4±1.1 | 53.8±0.8 | 42.2±0.8 |
| | + Greg | **54.1±2.2** | **64.1±1.3** | **60.6±1.5** | **45.1±0.8** |
| TP | NVDM | 32.0±0.3 | 31.5±0.2 | 30.8±0.7 | 29.4±0.7 |
| | + Greg | **40.0±0.7** | **38.9±0.6** | **38.0±1.4** | **33.7±1.1** |
| | PLDA | 27.0±0.2 | 37.4±0.6 | 30.6±0.4 | 26.3±0.0 |
| | + Greg | **30.5±0.7** | **41.7±0.6** | **32.3±0.6** | **26.5±0.1** |
| | SCHOLAR | 37.8±5.3 | 35.5±3.4 | 49.2±2.3 | 29.3±1.2 |
| | + Greg | **48.7±1.9** | **55.8±2.4** | **53.4±1.3** | **33.8±1.2** |
| | CLNTM | 38.2±3.3 | 33.1±1.1 | 45.9±2.0 | 34.1±1.5 |
| | + Greg | **42.2±3.9** | **58.0±1.3** | **53.9±2.0** | 34.1±2.5 |
| TN | NVDM | 3.5±0.1 | 3.6±0.1 | 3.0±0.2 | 2.3±0.2 |
| | + Greg | **9.7±0.4** | **8.9±0.2** | **8.1±1.0** | **4.7±0.3** |
| | PLDA | 1.5±0.1 | 6.8±0.4 | 3.1±0.2 | 1.1±0.0 |
| | + Greg | **3.3±0.2** | **9.6±0.6** | **4.5±0.3** | 1.1±0.0 |
| | SCHOLAR | 12.8±1.7 | 12.0±2.1 | 18.2±1.0 | 3.4±0.8 |
| | + Greg | **20.5±1.4** | **23.1±2.4** | **24.3±1.2** | **7.2±0.8** |
| | CLNTM | 8.5±2.1 | 6.6±1.9 | 17.0±0.9 | 6.7±0.7 |
| | + Greg | **14.4±3.5** | **24.2±0.5** | **23.8±1.5** | **8.1±1.5** |

## 5.2   Results

### 5.2.1   One Source to Different Targets

We set 20News as the source corpus and the other datasets as the target corpora for our experiments. The quality of the topical representation is measured by CA, TP and TN. The results for $K = 50$ are illustrated in Table 2, where the larger value between backbone and backbone with Greg under each setting is highlighted in boldface. From the results, it can be observed that applying Greg to various models significantly improves the CA, TP, and TN metrics across different target corpora in most instances. For example, by integrating NVDM with Greg, we improve CA from 39% to 63.7%, 38.8% to 60.2%, 31.2% to 55.8% and 73.2% to 80.6% on average for targets Webs, TMN, DBpedia and R8, respectively. Similarly, a large improvement can also be obtained for TP and TN after applying Greg. Overall, the results show that our approach effectively generalizes the neural topical representation across corpora in our experiments.

### 5.2.2   Different Sources to One Target

We fix the target and use different source datasets to further investigate NTM's generalization ability. We use TMN as the target, then the rest datasets are set as the sources, respectively. The results for $K$ equals 50 are illustrated in Table 3. Notably, "**20News_T**" indicates the evaluation is conducted on target corpus TMN where the model is trained on source corpus 20News. Based on the results, we summarize the following observations: We significantly improve documents' topical representation in the target corpus when different source datasets are used. For example, in terms of CA, by integrating Greg with NVDM, the performance in the target TMN is increased from 38.4% to 59.6%, 44.4% to 63.4%, 39.5% to 59.4% and 33.5% to 43.3% when 20News, Webs, DBpedia and R8 are used as the source, respectively. Similar observations can be made when setting a different target corpus such as R8, with the results illustrated in Appendix D.1.

### 5.2.3   Target as Noisy Corpus

We challenge the models with noisy datasets as the target, where the model is trained on original source datasets but evaluated on their noisy versions (e.g., **Dataset_N**). The experimental results on the noisy

Table 4: Target as noisy corpus

| (%,↑) | Method | 20News_N | Webs_N | TMN_N | DBpedia_N | R8_N |
|---|---|---|---|---|---|---|
| CA | NVDM | 25.8±0.2 | 49.7±0.9 | 51.3±0.5 | 58.5±1.0 | 80.3±0.4 |
|  | + Greg | **28.0±0.4** | **60.1±0.8** | **57.5±0.9** | **66.1±0.8** | **83.5±1.0** |
|  | PLDA | 23.1±0.6 | 52.2±0.9 | 52.1±0.6 | 55.3±2.1 | 70.6±0.4 |
|  | + Greg | **25.3±0.6** | **55.4±0.4** | **54.1±0.7** | **58.2±1.1** | **73.0±0.9** |
|  | SCHOLAR | **43.2±0.7** | 73.6±2.9 | 66.1±0.9 | 82.7±1.4 | 87.3±0.9 |
|  | + Greg | 42.3±2.1 | **86.0±0.7** | **77.5±0.3** | **84.1±1.4** | **88.4±0.5** |
|  | CLNTM | **39.8±1.3** | 70.7±1.3 | 66.1±0.7 | 70.0±1.6 | 87.3±0.6 |
|  | + Greg | 39.5±1.8 | **87.1±0.6** | **77.2±0.7** | **82.2±1.0** | **87.7±0.8** |
| TP | NVDM | 14.4±0.4 | 30.4±0.5 | 33.1±0.8 | 24.7±1.2 | 66.3±1.1 |
|  | + Greg | **14.7±0.5** | **33.0±0.8** | **35.7±1.0** | **26.6±1.3** | **66.8±1.3** |
|  | PLDA | 21.6±0.5 | 50.3±0.6 | 50.8±0.8 | 58.2±0.6 | 64.7±0.6 |
|  | + Greg | **23.3±0.4** | **53.7±0.4** | **52.6±0.6** | **62.0±1.1** | **66.8±1.0** |
|  | SCHOLAR | **39.5±1.0** | 46.6±2.7 | 54.0±1.5 | 74.2±1.9 | **83.7±1.2** |
|  | + Greg | 37.7±1.3 | **83.1±1.7** | **76.7±0.9** | **86.4±0.8** | 83.0±0.6 |
|  | CLNTM | **37.4±1.1** | 46.7±2.4 | 54.6±2.9 | 59.0±4.0 | **80.9±1.9** |
|  | + Greg | 30.2±3.8 | **84.6±1.2** | **74.9±1.1** | **81.7±1.3** | 80.4±2.1 |
| TN | NVDM | 8.8±0.2 | 7.3±0.1 | 4.5±0.2 | 14.1±0.7 | 16.5±0.6 |
|  | + Greg | **9.0±0.4** | **9.1±0.2** | **6.3±0.4** | **17.3±0.8** | **17.3±1.0** |
|  | PLDA | 10.7±0.4 | 18.3±0.4 | 14.0±0.1 | 35.8±0.5 | 15.1±0.4 |
|  | + Greg | **12.7±0.4** | **21.1±0.5** | **15.4±0.4** | **40.0±0.7** | **16.6±0.3** |
|  | SCHOLAR | 31.6±0.9 | 29.6±3.5 | 31.7±1.8 | **69.1±1.4** | 36.4±1.3 |
|  | + Greg | **32.0±1.1** | **53.2±1.4** | **39.1±0.4** | 68.5±1.1 | **37.3±0.8** |
|  | CLNTM | **30.4±1.4** | 28.8±2.0 | 33.3±1.9 | 54.5±3.2 | 34.3±1.3 |
|  | + Greg | 26.1±2.3 | **55.1±1.4** | **38.4±1.2** | **64.9±0.9** | **36.2±1.8** |

Table 5: Source corpus performance

| (%,↑) | Method | 20News | Webs | TMN | DBpedia | R8 |
|---|---|---|---|---|---|---|
| CA | NVDM | 40.1±0.2 | 60.1±0.8 | 61.9±0.8 | 74.5±0.6 | 87.6±0.6 |
|  | + Greg | **41.9±0.4** | **69.5±0.4** | **67.1±1.1** | **79.3±0.5** | **88.7±0.4** |
|  | PLDA | 35.2±0.8 | 62.0±1.3 | 62.7±0.8 | 69.5±1.1 | 79.7±0.5 |
|  | + Greg | **37.7±0.5** | **65.3±1.1** | **64.5±1.1** | **72.1±1.9** | **81.5±0.6** |
|  | SCHOLAR | **56.3±0.8** | 81.2±2.2 | 73.0±0.5 | 89.1±1.2 | 91.4±0.4 |
|  | + Greg | 54.1±2.2 | **88.1±0.9** | **82.6±0.5** | **91.0±1.1** | **93.3±0.4** |
|  | CLNTM | **55.6±1.0** | 77.9±2.0 | 72.6±0.7 | 76.2±1.3 | 92.1±0.4 |
|  | + Greg | 53.0±1.3 | **91.1±0.7** | **82.0±0.4** | **88.8±0.7** | **92.9±0.3** |
| TP | NVDM | **19.7±0.4** | 32.2±0.7 | 36.0±0.6 | 30.0±1.2 | 70.2±1.6 |
|  | + Greg | 19.1±0.5 | **34.4±0.7** | **38.2±1.1** | **31.5±1.3** | **70.3±1.7** |
|  | PLDA | 33.6±0.7 | 60.5±1.3 | 62.5±0.9 | 74.5±1.1 | 74.7±0.2 |
|  | + Greg | **35.8±0.5** | **63.9±0.9** | **63.8±1.1** | **77.0±0.7** | **77.2±0.9** |
|  | SCHOLAR | **54.6±0.8** | 51.6±3.1 | 58.5±1.6 | 80.1±3.0 | **91.1±0.6** |
|  | + Greg | 51.0±1.8 | **85.4±1.7** | **81.0±0.8** | **91.5±1.1** | 88.9±0.9 |
|  | CLNTM | **57.5±1.0** | 51.2±3.3 | 59.7±2.5 | 63.5±4.2 | **91.1±1.7** |
|  | + Greg | 44.5±5.0 | **88.8±0.9** | **79.7±1.1** | **88.2±1.4** | 90.0±1.5 |
| TN | NVDM | **14.2±0.4** | 8.3±0.4 | 6.5±0.3 | 20.3±1.1 | **22.0±0.7** |
|  | + Greg | 14.0±0.5 | **10.2±0.3** | **8.2±0.4** | **23.3±1.1** | 21.7±0.7 |
|  | PLDA | 22.3±0.3 | 27.4±1.0 | 23.6±0.3 | 54.2±0.4 | 24.0±0.4 |
|  | + Greg | **25.4±0.4** | **30.7±0.9** | **25.1±0.6** | **57.2±0.4** | **26.4±0.7** |
|  | SCHOLAR | **48.1±0.7** | 34.2±3.6 | 37.8±1.2 | **78.4±1.6** | 43.2±0.6 |
|  | + Greg | 46.7±1.7 | **55.8±1.5** | **44.5±0.4** | 75.3±1.3 | **44.8±1.2** |
|  | CLNTM | **49.1±0.3** | 34.7±2.5 | 39.1±1.8 | 63.1±3.4 | 43.7±1.4 |
|  | + Greg | 42.5±3.4 | **60.7±1.2** | **43.4±1.1** | **72.6±0.7** | **45.3±1.8** |

Table 6: Effect of DA on Greg

| (%,↑) | Method | Targets | | | |
|---|---|---|---|---|---|
|  |  | Webs | TMN | DBpedia | R8 |
| CA | Random to Similar | 62.7±0.4 | 58.3±0.2 | **56.0±0.7** | 80.5±0.5 |
|  | Highest to Similar | **63.7±0.3** | **60.2±0.4** | 55.8±0.4 | **80.6±0.5** |
|  | Lowest to Similar | 55.5±0.7 | 52.8±0.8 | 49.6±0.8 | 78.9±0.8 |
|  | Both to Similar | 60.7±1.2 | 57.3±0.1 | 52.4±1.5 | 79.3±0.9 |
| TP | Random to Similar | 34.7±1.0 | 39.6±1.6 | 25.5±0.9 | 63.5±1.8 |
|  | Highest to Similar | **35.8±1.5** | **40.1±1.5** | **26.3±0.9** | **63.7±1.8** |
|  | Lowest to Similar | 33.6±0.7 | 37.4±1.2 | 23.9±1.1 | 62.6±1.7 |
|  | Both to Similar | 34.8±1.2 | 38.6±1.7 | 25.5±1.2 | 62.4±2.1 |
| TN | Random to Similar | 12.5±1.0 | 9.1±0.7 | 15.3±1.0 | 15.3±1.0 |
|  | Highest to Similar | **12.7±0.8** | **9.6±0.8** | **15.7±0.5** | **15.5±0.7** |
|  | Lowest to Similar | 10.8±0.3 | 7.4±0.5 | 13.8±0.5 | 14.5±1.3 |
|  | Both to Similar | 11.9±0.9 | 8.4±0.9 | 15.0±0.9 | 14.9±0.9 |

Table 7: Effect of different distances on Greg

| (%,↑) | Method | Targets | | | |
|---|---|---|---|---|---|
|  |  | Webs | TMN | DBpedia | R8 |
| CA | Euclidean Distance | 41.1±1.1 | 39.7±0.2 | 38.0±0.9 | 69.3±0.6 |
|  | Cosine Distance | 24.1±0.3 | 26.9±0.2 | 14.6±0.3 | 59.5±0.6 |
|  | Hellinger Distance | 38.8±1.0 | 38.8±0.6 | 31.3±0.6 | 73.3±0.9 |
|  | TopicalOT | **63.7±0.3** | **60.2±0.4** | **55.8±0.4** | **80.6±0.5** |
| TP | Euclidean Distance | 30.0±0.4 | 32.1±0.6 | 21.0±0.5 | 61.5±1.3 |
|  | Cosine Distance | 23.4±0.1 | 25.8±0.1 | 11.6±0.4 | 55.3±0.6 |
|  | Hellinger Distance | 28.8±0.4 | 32.3±0.5 | 19.6±0.2 | 62.0±2.0 |
|  | TopicalOT | **35.8±1.5** | **40.1±1.5** | **26.3±0.9** | **63.7±1.8** |
| TN | Euclidean Distance | 6.5±0.2 | 3.8±0.2 | 9.9±0.5 | 12.4±1.1 |
|  | Cosine Distance | 4.3±0.1 | 2.2±0.2 | 5.4±0.3 | 10.0±0.4 |
|  | Hellinger Distance | 6.5±0.3 | 3.7±0.2 | 9.0±0.4 | 12.9±1.1 |
|  | TopicalOT | **12.7±0.8** | **9.6±0.8** | **15.7±0.5** | **15.5±0.7** |

targets at $K = 50$ are illustrated in Table 4. From this set of experiments, we can observe that (i) Greg continues showing its benefits in generalizing neural topical representation when the target domain contains noise in most settings, demonstrating improved robustness at the same time. (ii) It is noticeable that Greg causes a performance drop when applied to CLNTM on the 20News dataset. The potential reason is that CLNTM employs a contrastive learning approach to regularize topical representations alongside Greg, making it challenging to achieve a balance between the two regularizers during training. A further analysis of Greg under different settings is provided in Appendix D.2.

### 5.2.4 Source Corpus Performance

We illustrate the performance on the original corpus when applying Greg to different NTMs in Table 5. Based on these results, it can be observed that (i) Greg can improve the topical representation quality of the source documents at the same time for short documents (i.e., Webs, TMN and DBpedia) under most settings. (ii) For long document corpora like 20News and R8, there are instances where Greg leads to a performance drop on the source corpus. However, we believe that the advantages of using Greg are substantial, and that performance can be further improved across specific datasets through adjusted hyperparameters. A further analysis of Greg under different settings is provided in Appendix D.2.

### 5.3 Ablation Study and Hyperparameter Sensitivity

#### 5.3.1 Effect of Positive DA on Greg

Different DAs can be applied for Greg as long as it creates positive (e.g., similar) documents, which shows the flexibility of our framework. Here, we demonstrate our choice of DA by applying different positive DAs to Greg. Their resulting topical representation quality of target corpora is illustrated in Table 6, where the source corpus is 20News, and the backbone is NVDM with $K = 50$. Based on these results, "Highest to Similar" obtains the highest quality in almost all the target domains, which shows the best generalization capability among the DAs investigated. Thus, we apply "Highest to Similar" in Greg throughout our experiments.

#### 5.3.2 Effect of Distance Metrics on Greg

Here, we demonstrate the effectiveness of TopicalOT in Greg in enhancing NTMs' generalization power by changing different distance metrics. We consider other standard distances for the experiments, including Euclidean, Cosine and Hellinger distances. Their resulting topical representation quality of target corpora is illustrated in Table 7, where the source corpus is 20News, and the backbone is NVDM with $K = 50$. Based on these results, TopicalOT brings a significant improvement to the performance of the target domains and leaves a large margin compared to other distance metrics. It demonstrates the effectiveness of TopicalOT within our generalization regularization framework.

#### 5.3.3 Sensitivity to K in Short Document

Setting the number of topics (i.e., $K$) for topic models is one of the challenges in short-text topic modeling (Xuan et al., 2016; Qiang et al., 2020). From the experiments in previous sections, it can be noticed that Greg brings a huge improvement of topical representation quality for short documents when applying to SCHOLAR and CLNTM for most settings. Here, we explore how sensitive this improvement is to the settings of $K$. We plot the results for two short document corpora, Webs and TMN, in Figure 2. It can be observed that the topical representation quality of both SCHOLAR and CLNTM drops rapidly as $K$ increases, which indicates that they are sensitive to $K$ for short documents. By applying Greg, both SCHOLAR and CLNTM exhibit improved and stable topical representation quality across various settings of $K$. This indicates that Greg effectively addresses their sensitivity issues.

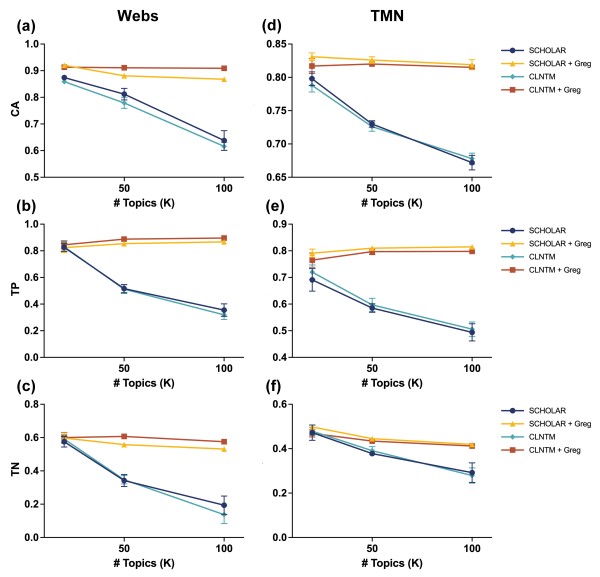

Figure 2: Effect of number of topics (i.e., $K$) to backbones and Greg.

## 6 Conclusion

In this work, we propose a new regularization loss that can be integrated into many existing neural topic models (NTMs) for training on one dataset and generalizing their topical representations to unseen documents without retraining. Our proposed loss, Greg, encourages NTMs to produce similar latent distributions for similar documents. The distance between document representations is measured by TopicalOT, which incorporates semantic information from both topics and words. Extensive experiments demonstrate that our framework, as a model-agnostic plugin for existing NTMs, significantly improves the generalization ability of NTMs. In the future, we believe that topic model generalization can be extended to generalizing both document representations and topics across different languages and modalities.

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

# A  Summary of Math Notations

Table A1: Mathematical notations and descriptions

| Category | Notation | Description |
|---|---|---|
| NTM | $K, V, I, B$ | Number of topics, vocabulary words, topic top words, batch size; |
| | $\boldsymbol{d}, \boldsymbol{x}, \boldsymbol{x}^s, \boldsymbol{x}^{aug}$ | Document, BOW vector, BOW of source document and its augmentation; |
| | $\boldsymbol{z}, \boldsymbol{z}^s, \boldsymbol{z}^{aug}$ | Topical representation, topical representation of source document and its augmentation; |
| | $\boldsymbol{X}, \boldsymbol{X}^s, \boldsymbol{X}^{aug}$ | Batch of BOW vectors, Batch of BOW of source documents and their augmentations; |
| | $\boldsymbol{Z}, \boldsymbol{Z}^s, \boldsymbol{Z}^{aug}$ | Batch of topical representations, Batch of topical representations of source documents and their augmentations; |
| | $\mathcal{D}, \mathcal{D}^S, \mathcal{D}^T$ | Text corpus, the source and target corpus; |
| | $\mathcal{V}, \mathcal{V}^S, \mathcal{V}^T$ | Vocabulary, source and target vocabulary; |
| | $\boldsymbol{T}, \tilde{\boldsymbol{T}}, \boldsymbol{t}_k, \tilde{\boldsymbol{t}}_k$ | Set of topics, set of estimated topics, topic $k$, and estimated topic $k$; |
| | $\theta, \phi, W$ | Encoder network, decoder network, and decoder weight; |
| OT | $\boldsymbol{X}, \boldsymbol{Y}$ | Supports of two discrete distributions; |
| | $\boldsymbol{a}, \boldsymbol{b}$ | Probability vectors; |
| | $\Delta^M$ | $M$-dimensional probability simplex; |
| | $D, D_{\boldsymbol{M}}, D_{\boldsymbol{M}, \lambda}$ | General distance, OT distance, Sinkhorn distance; |
| | $\boldsymbol{M}, \boldsymbol{M^t}, \boldsymbol{M^d}$ | General cost matrix, topic cost matrix, document cost matrix; |
| | $\boldsymbol{P}, \boldsymbol{U}$ | Transport matrix, transport polytope; |
| General | $\boldsymbol{E}, \boldsymbol{e}, L$ | Word embedding matrix, word embedding vector, embedding dimension; |
| | $\beta, \gamma, \lambda$ | Augmentation strength, regularization weight, Sinkhorn hyperparameter; |
| | $\mathcal{F}, f_N, f_I$ | Data augmentation function, normalizing function, function return top $I$ elements |

# B  Algorithm

---
**Algorithm 1** Neural Topic Model with Greg
---

Input: Dataset $\mathcal{D}^s = \{\boldsymbol{x}_i\}_{i=1}^N$, pre-trained word embeddings $\boldsymbol{E}$, topic number K, regularization weight $\gamma$, augmentation strength $\beta$

Output: $\theta, \phi$

 1: Randomly initialize $\theta$ and $\phi$;
 2: **while** not converged **do**
 3:     Sample a batch of data $\boldsymbol{X}$;
 4:     Compute $\boldsymbol{Z} = \text{softmax}(\theta(\boldsymbol{X}))$;
 5:     Compute $\boldsymbol{Z}^{aug}$ by Eq. (13);
 6:     Get topics $\tilde{\boldsymbol{T}}$ from $\phi$ by Eq. (8) and (10);
 7:     **for** each topic pairs $\tilde{\boldsymbol{t}}_{k_1}, \tilde{\boldsymbol{t}}_{k_2}$ **do**
 8:         Construct $\boldsymbol{M}^{\tilde{\boldsymbol{t}}_{k_1}, \tilde{\boldsymbol{t}}_{k_2}}$ in Eq. (12c);
 9:         Compute $\boldsymbol{M}^{\boldsymbol{d}}_{k_1, k_2}$ in Eq. (12b);
10:     **end for**
11:     Compute the loss defined in Eq. (14);
12:     Compute gradients w.r.t $\theta$ and $\phi$;
13:     Update $\theta$ and $\phi$ based on the gradients;
14: **end while**

## C  Datasets Statistics

Table C1: Statistics of the datasets

| Dataset | # Docs | Voc Size | Avg. Length | # Labels |
|---------|--------|----------|-------------|----------|
| 20News  | 18846  | 1997     | 87          | 20       |
| R8      | 7674   | 5047     | 56          | 8        |
| Webs    | 12337  | 4523     | 14          | 8        |
| TMN     | 32597  | 12004    | 18          | 7        |
| DBpedia | 19993  | 9830     | 23          | 14       |

## D  More Results

### D.1  Different Sources to One Target (R8)

Table D1: Different sources to one target (R8)

| (%,↑) | Method | Target (R8) | | | |
|-------|--------|-----------|---------|--------|-----------|
|       |        | 20News_R  | Webs_R  | TMN_R  | DBpedia_R |
| CA | NVDM | 72.6±0.3 | 73.8±0.9 | 78.6±0.6 | 72.2±0.6 |
|    | + Greg | **80.8±1.4** | **80.1±1.0** | **83.7±0.6** | **73.8±0.9** |
|    | PLDA | 59.7±0.3 | 63.5±0.9 | 74.9±1.2 | 63.6±0.6 |
|    | + Greg | **63.4±0.7** | **64.4±0.8** | **75.9±0.7** | **63.9±0.6** |
|    | SCHOLAR | **77.8±1.6** | 76.8±1.1 | 76.5±1.5 | **76.0±0.2** |
|    | + Greg | 75.2±1.5 | **77.6±0.7** | **84.2±0.9** | 75.1±0.8 |
|    | CLNTM | 74.0±1.6 | 75.9±1.3 | 76.6±1.0 | 73.5±0.6 |
|    | + Greg | **75.0±1.5** | **77.5±1.1** | **81.9±2.3** | **75.5±0.6** |
| TP | NVDM | 61.2±0.5 | 62.2±1.1 | 61.0±1.5 | **61.9±2.0** |
|    | + Greg | **65.1±2.9** | **64.8±2.6** | **64.7±1.5** | 61.8±2.2 |
|    | PLDA | 56.7±0.4 | **58.2±0.7** | 64.5±2.6 | 61.1±0.7 |
|    | + Greg | **57.8±1.1** | 56.7±0.7 | **68.5±2.3** | **62.1±0.9** |
|    | SCHOLAR | 61.6±1.5 | 64.3±0.6 | 58.8±3.4 | **64.1±0.6** |
|    | + Greg | **63.0±2.0** | **64.4±1.4** | **70.6±5.3** | 63.9±2.1 |
|    | CLNTM | **63.0±2.2** | **65.0±2.1** | 58.8±0.8 | **64.8±1.4** |
|    | + Greg | 62.8±2.1 | 64.3±1.4 | **68.9±5.4** | 62.5±2.3 |
| TN | NVDM | 12.9±0.5 | 13.4±0.8 | 13.2±0.6 | 12.6±0.9 |
|    | + Greg | **17.5±1.5** | **17.5±2.3** | **16.9±1.4** | **13.1±1.1** |
|    | PLDA | 7.8±0.3 | **10.6±0.9** | 20.0±1.7 | 11.8±0.2 |
|    | + Greg | **8.7±0.4** | 10.5±0.7 | **23.2±2.5** | **12.4±0.7** |
|    | SCHOLAR | 14.7±1.2 | 20.6±1.9 | 12.4±2.4 | **20.2±0.6** |
|    | + Greg | **16.2±2.2** | **20.8±1.2** | **35.4±5.0** | 18.9±1.2 |
|    | CLNTM | 15.0±1.2 | 18.8±1.9 | 13.4±2.0 | **20.3±1.2** |
|    | + Greg | 15.0±1.6 | **19.7±1.7** | **34.2±4.8** | 18.0±0.8 |

Here, we conduct similar experiments to those described in Section 5.2.2, where the target corpus is changed to R8. The results are illustrated in Table D1. Notably, "**20News_R**" indicates the evaluation is conducted

on target corpus R8 where the model is trained on source corpus 20News. Based on the results, we observe that Greg brings improvements to the topical representation of the target corpus R8 under most settings when a different source corpus is used.

## D.2 Significance Test

Table D2: P-value of paired T-test on target corpus ($\alpha = 0.05$)

| Metric | Target | | | |
|---|---|---|---|---|
| | Webs | TMN | DBpedia | R8 |
| CA | **3.66e-12** | **6.49e-70** | **4.70e-09** | **1.25e-25** |
| TP | **4.88e-07** | **1.68e-43** | **7.35e-05** | **9.71e-11** |
| TN | **1.31e-09** | **5.95e-55** | **1.87e-05** | **1.23e-12** |

Table D3: P-value of paired T-test on noisy corpus ($\alpha = 0.05$)

| Metric | Noisy Target | | | | |
|---|---|---|---|---|---|
| | Webs | TMN | DBpedia | 20News | R8 |
| CA | **1.42e-12** | **3.79e-14** | **1.21e-10** | 7.01e-02 | **2.67e-09** |
| TP | **2.98e-07** | **2.25e-10** | **9.90e-07** | 7.45e-01 | 4.85e-01 |
| TN | **1.11e-07** | **8.03e-10** | **4.44e-07** | 6.37e-01 | **1.30e-04** |

Table D4: P-value of paired T-test on source corpus ($\alpha = 0.05$)

| Metric | Source | | | | |
|---|---|---|---|---|---|
| | Webs | TMN | DBpedia | 20News | R8 |
| CA | **2.24e-12** | **5.12e-14** | **5.53e-11** | 3.42e-01 | **2.96e-16** |
| TP | **1.67e-07** | **1.43e-09** | **1.23e-06** | **3.10e-05** | 5.61e-01 |
| TN | **8.88e-08** | **3.54e-08** | **3.62e-06** | **2.23e-02** | **1.20e-03** |

We conduct paired t-tests across our previous experimental results to demonstrate Greg's performance across different scenarios from a general view. We collect paired differences in performance metrics between the original model and the model incorporating Greg across various datasets under the following settings: (i) different target corpora (Table D2), (ii) noisy versions of the source corpora (Table D3), and (iii) the original source corpora (Table D4). We set the significance level (i.e., $\alpha$) as 0.05 for all paired t-tests. The P-value lower than $\alpha$ is highlighted in boldface in tables, indicating a significant difference between the model with and without Greg. We have the following observations based on the results: (i) From Table D2, the benefit of Greg that improves the performance of different target corpora is significant for all targets. (ii) From Table D3, when the target is a noisy corpus, the improvement by Greg is significant for all short corpora (e.g., Webs, TMN and DBpedia). For the long-document noisy target such as 20News, Greg shows comparable performance with the original model. (iii) From Table D4, the improvement to source corpus performance by Greg is significant for most settings.

### D.3 Effect of DA on Topical Representation

Table D5: Effect of different DA on topical representation

| Dataset | Method | CD (%, ↓) | | | HD (%, ↓) | | | TopicalOT (%, ↓) | | |
|---|---|---|---|---|---|---|---|---|---|---|
| | | LDA | NVDM | NVDM + Greg | LDA | NVDM | NVDM + Greg | LDA | NVDM | NVDM + Greg |
| 20News | Random Drop | 13.3±0.4 | 9.8±0.1 | **7.3±0.1** | 33.3±0.4 | 15.8±0.1 | **13.6±0.1** | 10.8±0.2 | 8.9±0.1 | **3.8±0.1** |
| | Random Insertion | 14.2±0.4 | 9.7±0.1 | **7.0±0.1** | 38.5±0.4 | 15.7±0.1 | **13.2±0.1** | 13.2±0.2 | 8.8±0.0 | **3.6±0.1** |
| | Random to Similar | 15.8±0.3 | 11.1±0.1 | **7.2±0.1** | 37.6±0.3 | 16.9±0.1 | **13.6±0.1** | 12.5±0.2 | 9.5±0.0 | **3.7±0.1** |
| | Highest to Similar | 17.6±0.5 | 11.3±0.1 | **7.1±0.1** | 38.8±0.4 | 17.0±0.1 | **13.4±0.1** | 12.7±0.3 | 9.5±0.0 | **3.7±0.1** |
| | Lowest to Similar | 17.6±0.1 | 10.2±0.1 | **7.4±0.1** | 39.6±0.3 | 16.1±0.1 | **13.6±0.1** | 14.2±0.2 | 9.0±0.0 | **3.8±0.1** |
| | Random to Dissimilar | 26.3±1.6 | 20.7±1.1 | **15.0±0.5** | 47.7±1.0 | 23.7±0.7 | **19.9±0.4** | 18.4±0.4 | 13.2±0.4 | **5.5±0.1** |
| | Highest to Dissimilar | 32.4±2.0 | 21.2±1.0 | **15.0±0.4** | 51.5±1.1 | 23.9±0.6 | **19.9±0.3** | 20.8±0.5 | 13.4±0.4 | **5.5±0.1** |
| | Lowest to Dissimilar | 20.4±1.1 | 19.5±1.2 | **14.7±0.4** | 44.0±0.9 | 22.8±0.7 | **19.6±0.3** | 16.0±0.4 | 12.8±0.4 | **5.4±0.1** |
| Webs | Random Drop | 5.9±0.2 | 10.0±0.1 | **5.6±0.1** | 20.2±0.3 | 16.5±0.0 | **12.3±0.1** | 8.7±0.1 | 8.8±0.1 | **3.1±0.1** |
| | Random Insertion | 4.8±0.2 | 9.4±0.1 | **4.7±0.1** | 23.6±0.3 | 15.8±0.1 | **11.0±0.1** | 9.3±0.1 | 8.4±0.1 | **2.8±0.1** |
| | Random to Similar | 5.6±0.4 | 10.8±0.1 | **5.0±0.1** | 21.4±0.5 | 17.1±0.1 | **11.4±0.1** | 8.4±0.2 | 9.0±0.2 | **2.9±0.1** |
| | Highest to Similar | 5.8±0.4 | 10.3±0.2 | **4.7±0.1** | 21.1±0.4 | 16.7±0.1 | **11.2±0.1** | 8.3±0.2 | 8.9±0.2 | **2.9±0.1** |
| | Lowest to Similar | 4.8±0.3 | 11.0±0.1 | **5.3±0.1** | 21.2±0.4 | 17.2±0.1 | **11.7±0.1** | 8.2±0.1 | 9.1±0.2 | **3.0±0.1** |
| | Random to Dissimilar | 8.8±0.3 | 11.9±0.2 | **6.4±0.1** | 29.6±0.4 | 18.1±0.2 | **13.0±0.1** | 12.6±0.2 | 9.6±0.2 | **3.3±0.1** |
| | Highest to Dissimilar | 8.4±0.3 | 11.3±0.1 | **6.1±0.1** | 28.9±0.4 | 17.7±0.1 | **12.8±0.1** | 12.5±0.2 | 9.4±0.1 | **3.2±0.1** |
| | Lowest to Dissimilar | 7.7±0.4 | 11.9±0.3 | **6.6±0.2** | 27.7±0.7 | 18.0±0.2 | **13.2±0.2** | 11.5±0.3 | 9.6±0.1 | **3.4±0.1** |

**Setup** We study the effect of different DAs on topical representations. As most topic models work with BOWs, we focus on the word-level DAs described in Table 1. The effect of DAs on topical representation is measured by the distance between the topical representations of original documents and augmentations. Specifically, we use the trained model to infer the topical representation of the test documents and their augmentations; Then different distance metrics are applied to calculate the distance between the topical representations of a document and its augmentations. The choices of distance metrics include Cosine distance (CD), Hellinger distance (HD) and TopicalOT. The choice of models here includes LDA, NVDM and NVDM with Greg; Moreover, we train these models with $K = 50$ on one long document corpus 20News and one short document corpus Webs; For the settings of DA, the augmentation strength is set as 0.5. For approaches based on word similarities, the number of top similar/dissimilar words considered for replacement is 20, where the Cosine similarity between GloVe word embeddings is applied to provide word similarity.

**Result** From the results in Table D5, we have the following observations: (i) When more words are perturbed (e.g., in long documents of 20News), NTMs such as NVDM are more stable (i.e., lower distances obtained) to DAs than probabilistic topic models such as LDA. While in the case that fewer words are perturbed, such as in Webs, LDA is more stable than NVDM. (ii) DAs that replace with similar words bring less effect than those that replace with dissimilar words. (iii) Interestingly, adding noise by random drop or insertion has a similar effect to replacing with similar words in our settings. These observations are cues of the intrinsic generalization ability of NTMs, which is further enhanced by Greg in this work.

### D.4 Hyperparameter Sensitivity of Greg

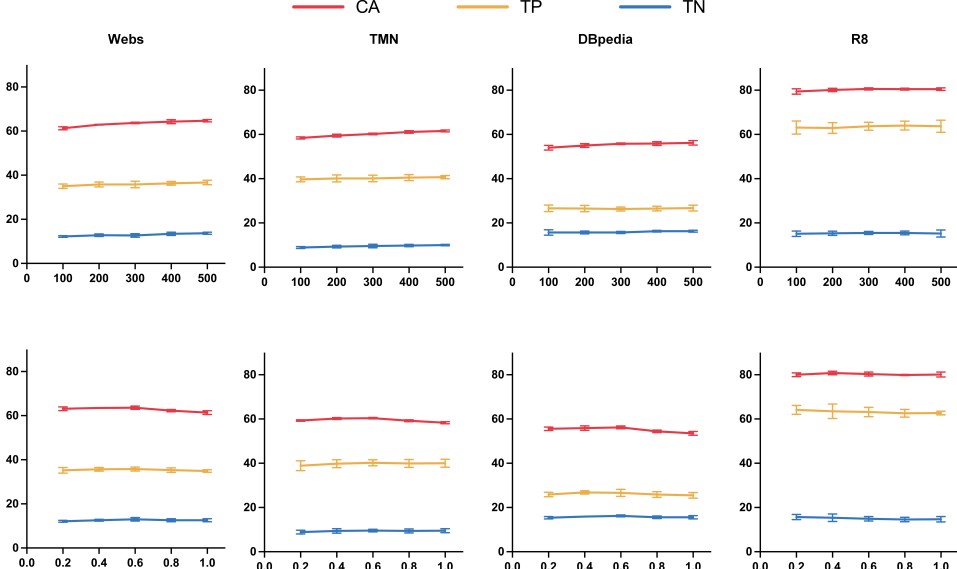

Figure D1: Hyperparameter sensitivity of Greg. The x-axis of the first row is the regularization weight $\gamma$; The x-axis of the second row is the augmentation rate $\beta$.

Here, we study the sensitivity to the setting of hyperparameters in Greg, focusing on the regularization weight $\gamma$ and augmentation weight $\beta$. We attach Greg to NVDM at $K = 50$ on the "One Source to Different Targets" tasks as in Section 5.2.1. We vary the regularization weight and augmentation rate, respectively, and record the performance of different metrics on different target corpora. Again, experiments are conducted 5 times with different random seeds. As shown in Figure D1, whether varying the regularization weight or the augmentation rate within a wide range, the benefits of Greg to target corpora performance still hold, and with little influence. It demonstrates Greg is not sensitive to the setting of its hyperparameters, thus its benefits are general, robust and reliable.

### D.5  Qualitative Analysis of Topics

Table D6: Example topics

| Topic | Top 10 Words (first rows: PLDA, second rows: PLDA + Greg) |
|---|---|
| ecosystems | habitat endemic tropical natural subtropical forest threatened loss moist ecuador |
| | habitat endemic natural tropical forest subtropical loss moist threatened ecuador |
| snail | snail marine gastropod sea mollusk family specie slug land terrestrial |
| | snail sea gastropod marine mollusk family specie genus land terrestrial |
| location | village mi east county km voivodeship central lie gmina poland |
| | village county district population central mi gmina administrative voivodeship poland |
| insects | moth wingspan family larva feed mm noctuidae tortricidae geometridae arctiidae |
| | moth described arctiidae geometridae noctuidae family snout subfamily beetle genus |
| religious | st church england century mary catholic saint parish paul roman |
| | church st historic catholic parish street saint england mary place |
| music | album released music singer song band record musician songwriter rock |
| | album released music record studio song singer band single debut |
| river | river long flow km mile tributary lake near creek source |
| | river tributary long near mile flow km basin source creek |

Although our primary focus is the generalization of document representation, we show examples of the learned topics to understand what topics are captured after integrating Greg. We choose the learned topics on DBpedia using the backbone PLDA. We pick the top coherent topics learned by PLDA and find their alignments in the learned topics of PLDA with Greg to explore the difference. The results are shown in Table D6. From the "location" topic, we observe that the top words are more coherent and related to the administrative division after using Greg; Within the "music" topic, Greg can identify words such as "studio", which may diversify the range of music-related words.

