# OpenReview forum: "Towards Generalizing Neural Topical Representations"
_TMLR — Rejected by TMLR_

### Review · Reviewer_iWtW · 2025-03-29

**Summary Of Contributions:**

The paper presents a new loss paradigm (GREG) applicably to most neural topic modeling backbones. It aims to generalize topical representations such that topic models better generalize to unseen documents. This is achieved by minimizing the Topical Optimal Transport (TopicalOT) distance between topical representations of similar documents. For each training document, a similar document is obtained by augmentation.

**Audience:**

Yes

**Claims And Evidence:**

Yes

**Requested Changes:**

# Changes and Questions

- Could you please include conventional topic quality metrics and compare with and without GREG [1]. In addition, since topic evaluation is rather difficult include newer metrics such as [2, 3]
- How do non neural models compare without implementing GREG, e.g. standard LDA, or BERTopic
- While you already include a few models, it would be interesting to see some of the newer, better performing models including GREG. E.g. some implemented here [4]
- How does GREG compare to other optimal transport modeling ideas, such as FASTopic [5]
- How does it compare to few-shot classification models such as [6]. The core idea of generalizing to unseen data seems to be similar.
- What about LLM based models such as [7, 8]


---
[1] Lau, Jey Han, David Newman, and Timothy Baldwin. "Machine reading tea leaves: Automatically evaluating topic coherence and topic model quality." Proceedings of the 14th Conference of the European Chapter of the Association for Computational Linguistics. 2014.
[2] Thielmann, Anton, et al. "Topics in the haystack: Enhancing topic quality through corpus expansion." Computational Linguistics 50.2 (2024): 619-655.
[3] Thielmann, Anton, et al. "STREAM: Simplified Topic Retrieval, Exploration, and Analysis Module." Proceedings of the 62nd Annual Meeting of the Association for Computational Linguistics (Volume 2: Short Papers). 2024.
[4] Wu, Xiaobao, Fengjun Pan, and Anh Tuan Luu. "Towards the TopMost: A topic modeling system toolkit." arXiv preprint arXiv:2309.06908 (2023).
[5] Wu, Xiaobao, et al. "FASTopic: Pretrained transformer is a fast, adaptive, stable, and transferable topic model." Advances in Neural Information Processing Systems 37 (2024): 84447-84481.
[6] Thielmann, Anton F., Christoph Weisser, and Benjamin Säfken. "Human in the Loop: How to Effectively Create Coherent Topics by Manually Labeling Only a Few Documents per Class." Proceedings of the 2024 Joint International Conference on Computational Linguistics, Language Resources and Evaluation (LREC-COLING 2024). 2024.
[7] Reuter, Arik, et al. "Gptopic: Dynamic and interactive topic representations." arXiv preprint arXiv:2403.03628 (2024).
[8] Pham, Chau Minh, et al. "Topicgpt: A prompt-based topic modeling framework." arXiv preprint arXiv:2311.01449 (2023).

**Strengths And Weaknesses:**

# Strength
- The paper is overall well written
- Notation is very clear, consistent and concise
- The general idea is - to the best of my knowledge - novel

# Weaknesses
- Although already including a dedicated literature section, the paper is missing quite the large amount of recent innovations in the field (see below, as well as in *Requested changes* in the field after this one)
- The evaluation is missing quite a lot of depth. Most importantly, the overall topic quality with and without GREG is not really compared.
   - Additionally training times with and without GREG should be compared.
   - It should also be evaluated against non neural topic models to see how well they generalize
   - (See requested changes below)


---
[1] Reuter, Arik, et al. "Probabilistic Topic Modeling With Transformer Representations." IEEE Transactions on Neural Networks and Learning Systems (2025).
[2]. Wu, Xiaobao, Thong Nguyen, and Anh Tuan Luu. "A survey on neural topic models: methods, applications, and challenges." Artificial Intelligence Review 57.2 (2024): 18.
[3] Ke, Zheng Tracy, and Minzhe Wang. "Using SVD for topic modeling." Journal of the American Statistical Association 119.545 (2024): 434-449.
[4] Thielmann, Anton, et al. "STREAM: Simplified Topic Retrieval, Exploration, and Analysis Module." Proceedings of the 62nd Annual Meeting of the Association for Computational Linguistics (Volume 2: Short Papers). 2024.

---

> ### Author Response · Authors · 2025-04-24
> **Response to Reviewer iWtW**
>
> We sincerely thank you for your comments and constructive suggestions.
>
> &nbsp;
>
> Thank you for the suggestion to include more recent works as baselines, and for pointing out specific pieces of relevant research. Your recommendations are very helpful, and we will include these works in the related work section of the revised version. We have incorporated several new baselines into our experiments based on your suggestions. These include non-neural models, recent neural topic models, latest LLM-based topic models, and a model that incorporates Optimal Transport (LLM-ITL [1]). Where applicable, we have also applied Greg to these new baselines. For further details, please refer to our global response (first section).
>
> &nbsp;
>
> For the comparison of training times with Greg, please refer to our global response (second section).
>
> &nbsp;
>
> Although the topic quality is not the main focus of this work—as our primary objective is to improve the generalization capability in terms of document representation—we agree that topic quality remains an important component of a topic model. To address this, we include an evaluation of topic quality using a widely adopted metric, NPMI (Normalized Pointwise Mutual Information [2]), using the Palmetto package [3]. The results are presented as follows (NPMI values have been converted to percentages for better use of table space):
>
> **Table: Topic coherence (NPMI)**
> | Model     | 20News | Webs       | TMN        | DBpedia    | R8         |
> |-----------|------------|------------|------------|------------|------------|
> LDA | 12.5 ± 0.9 | 12.5 ± 1.3 | 15.1 ± 0.6 | 15.0 ± 0.7 | 5.2 ± 0.4 |
> NMF | 11.0 ± 0.7 | 11.3 ± 0.6 | 15.2 ± 0.6 | 17.1 ± 0.5 | 7.9 ± 0.6 |
> BERTopic | 11.3 ± 1.0 | 12.8 ± 0.5 | 15.9 ± 0.2 | 24.5 ± 0.6 | 11.5 ± 1.0|
> | NVDM | 0.6 ± 1.1 | 3.1 ± 0.7 | 1.7 ± 0.3 | 2.6 ± 1.3 | 1.1 ± 0.6 |
> | NVDM+ Greg | 2.7 ± 0.3 | 5.4 ± 0.6 | 2.4 ± 1.4 | 1.6 ± 1.2 | 1.6 ± 1.1 |
> | ProdLDA | 9.4 ± 0.9 | 11.8 ± 1.4 | 13.4 ± 0.8 | 18.2 ± 0.6 | 3.5 ± 1.1 |
> | ProdLDA  + Greg | 10.1 ± 0.9 | 12.0 ± 1.9 | 13.5 ± 0.4 | 18.6 ± 0.4 | 3.3 ± 1.5 |
> | SCHOLAR | 11.7 ± 0.4 | 9.4 ± 1.2 | 9.7 ± 1.3 | 15.4 ± 2.1 | 4.4 ± 1.1 |
> | SCHOLAR + Greg | 3.0 ± 1.6 | 7.8 ± 2.4 | 8.5 ± 1.6 | 10.3 ± 0.6 | 1.9 ± 1.3 |
> | CLNTM | 13.0 ± 0.6 | 9.8 ± 0.7 | 9.2 ± 1.0 | 13.7 ± 1.6 | 6.5 ± 1.0 |
> | CLNTM  + Greg | 4.8 ± 3.3 | 6.2 ± 1.0 | 13.9 ± 2.0 | 9.5 ± 2.4 | 2.6 ± 0.1 |
>
>
>
> From the results, we observe that applying Greg to NVDM and PLDA leads to an improvement in topic quality, whereas applying it to SCHOLAR or CLNTM results in a decrease in performance. A potential reason for this is that many recent neural topic models are specifically designed to optimize topic quality, while our regularization term is primarily targeted at improving document representation. This may introduce a trade-off between the two learning objectives (i.e., the original topic model and Greg). In practice, we recommend applying Greg based on the intended use case and conducting careful hyperparameter tuning (as we set the same set of hyper-parameters for Greg for all datasets and backbones) —such as adjusting the regularization weight—to balance the performance between different components. Moreover, improving generalization capability in terms of both document representations and topic quality would be an interesting direction for future exploration.
>
> &nbsp;
>
> [1] Yang, X., Zhao, H., Xu, W., Qi, Y., Lu, J., Phung, D., & Du, L. (2024). Neural Topic Modeling with Large Language Models in the Loop. arXiv preprint arXiv:2411.08534.
>
>
>
> [2] Lau, J. H., Newman, D., & Baldwin, T. (2014, April). Machine reading tea leaves: Automatically evaluating topic coherence and topic model quality. In Proceedings of the 14th Conference of the European Chapter of the Association for Computational Linguistics (pp. 530-539).
>
> [3] Röder, M., Both, A., & Hinneburg, A. (2015, February). Exploring the space of topic coherence measures. In Proceedings of the eighth ACM international conference on Web search and data mining (pp. 399-408).

---

> > ### Comment · Reviewer_iWtW · 2025-04-25
> > **Further Questions/Changes**
> >
> > Dear authors,
> >
> > thank you for your rebuttal and your answers.
> > I have read the general answer as well as the answers to the other reviewers.
> > While a lot of my questions have been addressed, there are still some open questions, especially regarding the additional results.
> >
> >  - As outlined in multiple papers [1, 2] NPMI is not the best metric to use, to assess topic quality. I ask you again to use the metrics introduced in [3], implemented in [4] for better human alignment.
> >  - I would ask again to kindly include the SOTA of OT topic models, FasTopic in your comparison [5]
> >  - I am generally quite curious about the results for NPMI that you presented, since it seems that LDA and NMF are very close to the best performing models. This does not align with recent Literature [3, 5, 6] and I would ask you to include your use hyperparameters and compare it against the results from recent Literature.
> >  - How does it compare to few-shot classification models such as [6]. The core idea of generalizing to unseen data seems to be similar.
> >
> > ---
> > [1] Hoyle, Alexander, et al. "Is automated topic model evaluation broken? the incoherence of coherence." Advances in neural information processing systems 34 (2021): 2018-2033.
> > [2] Khodorchenko, Maria, Nikolay Butakov, and Denis Nasonov. "Towards better evaluation of topic model quality." 2022 32nd Conference of Open Innovations Association (FRUCT). IEEE, 2022.
> > [3] Thielmann, Anton, et al. "Topics in the haystack: Enhancing topic quality through corpus expansion." Computational Linguistics 50.2 (2024): 619-655.
> > [4] Thielmann, Anton, et al. "STREAM: Simplified Topic Retrieval, Exploration, and Analysis Module." Proceedings of the 62nd Annual Meeting of the Association for Computational Linguistics (Volume 2: Short Papers). 2024.
> > [5] Wu, Xiaobao, et al. "FASTopic: Pretrained transformer is a fast, adaptive, stable, and transferable topic model." Advances in Neural Information Processing Systems 37 (2024): 84447-84481.
> > [6] Thielmann, Anton F., Christoph Weisser, and Benjamin Säfken. "Human in the Loop: How to Effectively Create Coherent Topics by Manually Labeling Only a Few Documents per Class." Proceedings of the 2024 Joint International Conference on Computational Linguistics, Language Resources and Evaluation (LREC-COLING 2024). 2024.

---

> > > ### Author Response · Authors · 2025-04-27
> > > **Reply**
> > >
> > > Dear Reviewer,
> > >
> > > We are doing our best to run the additional experiments you recommended, including the use of further evaluation metrics for topic quality, the inclusion of state-of-the-art baselines (Fastopic and Fastopic + Greg), as well as the few-shot classification model you suggested.
> > >
> > > As the rebuttal deadline is approaching, we are unsure whether additional comments can be submitted afterward. Therefore, we would like to provide a quick response here. If it is possible to add new comments after the deadline, we will share the new results at that time. Otherwise, we will include these findings in a future revision of the paper.
> > >
> > > Regarding the inconsistency in NPMI scores compared to some other works, several factors may contribute to this, including differences in data preprocessing (e.g., vocabulary construction, data splits), dataset size, and experimental settings (e.g., the number of topics K). This is particularly relevant for neural topic models, where performance may degrade under different settings without extensive hyperparameter tuning. Additionally, we report both topic quality and document representation quality using the same model checkpoint (i.e., evaluating both components at the same model state), which differs from approaches that select the best values for each component separately (e.g., reporting NPMI and ACC at different model states). For more discussion on the potential inconsistency between these two components, please refer to Figure 1 of [1].
> > >
> > > Thank you once again for your valuable suggestions—they have greatly strengthened our work!
> > >
> > > [1] Yang, X., Zhao, H., Phung, D., Buntine, W., & Du, L. (2024). Llm reading tea leaves: Automatically evaluating topic models with large language models. arXiv preprint arXiv:2406.09008.

---

> > > > ### Comment · Reviewer_iWtW · 2025-04-28
> > > > **Reply**
> > > >
> > > > Dear authors,
> > > >
> > > > thank you for your answer and your commitment. I would greatly appreciate it, if due to the inconsistencies in NPMI, you would also consider one of the other metrics mentioned [1].
> > > >
> > > > As for the end of the rebuttal period, the AC can see the discussions as well as make preliminary decisions based on required revisions.
> > > >
> > > >
> > > > ---
> > > > [1] Thielmann, Anton, et al. "Topics in the haystack: Enhancing topic quality through corpus expansion." Computational Linguistics 50.2 (2024): 619-655.

---

### Review · Reviewer_JJqT · 2025-04-06

**Summary Of Contributions:**

In this paper, the authors aim to improve the performance of neural topic models (i.e., NVDM, ProdLDA, SCHOLAR, and CLNTM in the experiments) so that their representation power for documents generalizes reliably across corpora. Inspired by the Hierarchical OT (HOT) distance from Yurochkin et al. (2019), the authors propose the Topical Optimal Transport (TopicalOT) distance to measure the semantic distance between each pair of documents. The paper is well motivated from the methodology perspective, but the experiments need to be enhanced as shown in weaknesses.

**Audience:**

Yes

**Claims And Evidence:**

No

**Requested Changes:**

- Although the results in Table 7 validate the effectiveness of TopicalOT within the proposed generalization regularization framework when compared with other distance metrics, can you describe the possible reason for the following statement? However, these distances (i.e., Euclidean, Cosine, and Hellinger distances) cannot sufficiently capture the semantics distance between documents in terms of their topic distributions.
- In Figure 2, the smallest number of topics should be plotted.
- Our main contributions are summarized as followings -> Our main contributions are summarized as follows.
- Some of the arXiv preprints in the references have been presented at conferences, e.g., 1) “Neural topic model via optimal transport,” and 2) “Topic modelling meets deep neural networks: a survey.” These references should be formatted by listing the published versions, if available.

**Strengths And Weaknesses:**

Strengths:
- This paper aims to investigate the generalization ability of neural topic models across corpora, which is a topic that deserves in-depth study in this area.
- The paper is written clearly and thus is easy to follow.

Weaknesses:
- In the experiments, the authors demonstrate the benefits of their generalization regularization (Greg) framework only when it is integrated into four backbones and subsequently compared with them, i.e., NVDM (Miao et al., 2017), ProdLDA (Srivastava & Sutton, 2017), SCHOLAR (Card et al., 2017), and CLNTM (Nguyen & Luu, 2021). In recent years, given that various newer neural topic models and large language model-based topic models have emerged, it is necessary to present their performance to validate that the generalization ability of state-of-the-art topic models is indeed insufficient across corpora.
- To alleviate the high space complexity issue, the authors reduce the dimension of each topic by considering only the top I words that have the largest weights in the topic. It is suggested that the authors conduct a sensitivity analysis on the hyperparameter I.
- The authors suppose a stochastic function can produce a random augmentation x^{aug} that is semantically similar to x^s. According to the approaches presented in Table 1, it is suggested that the authors provide evidence to support the above assumption one-by-one.

---

> ### Author Response · Authors · 2025-04-24
> **Response to Reviewer JJqT**
>
> We sincerely thank you for your comments and constructive suggestions.
>
> &nbsp;
>
> For Weaknesses 1, please refer to our global response.
>
> &nbsp;
>
> For Weakness 2, we conducted an additional sensitivity analysis on the hyperparameter $I$. Specifically, we varied the value of $I$ from 3 to 10 in Greg and applied it to NVDM on the task as in Table 2. The results are presented as follows:
>
> **Table: Hyperparameter study of $I$ (CA)**
> | | Webs       | TMN        | DBpedia    | R8         |
> |-----------|------------|------------|------------|------------|
> | $I=3$ | 62.5 ± 1.6 | 59.3 ± 0.3 | 54.7 ± 1.2 | 80.3 ± 1.6 |
> | $I=5$ | 63.1 ± 1.7 | 59.7 ± 1.2 | 54.9 ± 0.5 | 79.3 ± 0.3 |
> | $I=7$ | 63.8 ± 0.6 | 59.7 ± 0.5 | 55.8 ± 0.8 | 78.5 ± 2.0 |
> | $I=10$ | 63.7 ± 0.3 | 60.2 ± 0.4 | 55.8 ± 0.4 |80.6 ± 0.5 |
>
> &nbsp;
>
> **Table: Hyperparameter study of $I$ (TP)**
> | | Webs       | TMN        | DBpedia    | R8         |
> |-----------|------------|------------|------------|------------|
> | $I=3$ | 33.9 ± 0.5 | 39.9 ± 1.1 | 25.2 ± 1.1 | 61.7 ± 2.7 |
> | $I=5$ | 34.1 ± 0.3 | 40.1 ± 0.9 | 25.1 ± 2.4 | 61.3 ± 2.0 |
> | $I=7$ | 34.9 ± 1.3 | 39.2 ± 2.4 | 25.6 ± 1.4 | 61.0 ± 3.8 |
> | $I=10$ | 35.8 ± 1.5 | 40.1 ± 1.5 | 26.3 ± 0.9 | 63.7 ± 1.8 |
>
> &nbsp;
>
> **Table: Hyperparameter study of $I$ (TN)**
> | | Webs       | TMN        | DBpedia    | R8         |
> |-----------|------------|------------|------------|------------|
> | $I=3$ | 12.1 ± 0.1 | 8.7 ± 0.8 | 15.3 ± 0.2 | 14.8 ± 0.8 |
> | $I=5$ | 12.5 ± 0.2 | 8.9 ± 0.3 | 15.0 ± 0.7 | 14.2 ± 1.0 |
> | $I=7$ | 11.7 ± 0.4 | 8.7 ± 0.7 | 15.7 ± 0.5 | 14.0 ± 1.4 |
> | $I=10$ | 12.7 ± 0.8 | 9.6 ± 0.8 | 15.7 ± 0.5 | 15.5 ± 0.7 |
>
> From the results, we observe that varying the value of $I$ leads to slight differences in performance. However, we believe that the fluctuation remains within an acceptable range, indicating low sensitivity to this hyperparameter. In practice, we suggest that further hyperparameter tuning of $I$ based on the dataset and backbone may lead to improved performance.
>
> &nbsp;
>
> For Weakness 3, we investigate the effect of different word-level document augmentations (DAs) on topical representations, as detailed in Appendix D.3. Additionally, we conduct an ablation study to justify our choice of using “Highest to Similar” in Greg. Specifically, we apply different positive document augmentation strategies within Greg, using NVDM as the backbone model. The results are presented as follows:
>
> **Table: Effect of DAs on Greg (CA)**
> | Method | Webs       | TMN        | DBpedia    | R8         |
> |-----------|------------|------------|------------|------------|
> | Random to Similar | 62.7 ± 0.4 | 58.3 ± 0.2 | **56.0 ± 0.7** | 80.5 ± 0.5 |
> | Highest to Similar | **63.7 ± 0.3** | **60.2 ± 0.4** | 55.8 ± 0.4 | **80.6 ± 0.5** |
> | Lowest to Similar | 55.5 ± 0.7 | 52.8 ± 0.8 | 49.6 ± 0.8 | 78.9 ± 0.8 |
> | Both to Similar | 60.7 ± 1.2 | 57.3 ± 0.1 | 52.4 ± 1.5 | 79.3 ± 0.9 |
>
> &nbsp;
>
> **Table: Effect of DAs on Greg (TP)**
> | Method | Webs       | TMN        | DBpedia    | R8         |
> |-----------|------------|------------|------------|------------|
> | Random to Similar | 34.7 ± 1.0 | 39.6 ± 1.6 | 25.5 ± 0.9 | 63.5 ± 1.8 |
> | Highest to Similar | **35.8 ± 1.5** | **40.1 ± 1.5** | **26.3 ± 0.9** | **63.7 ± 1.8** |
> | Lowest to Similar | 33.6 ± 0.7 | 37.4 ± 1.2 | 23.9 ± 1.1 | 62.6 ± 1.7 |
> | Both to Similar | 34.8 ± 1.2 | 38.6 ± 1.7 | 25.5 ± 1.2 | 62.4 ± 2.1 |
>
> &nbsp;
>
> **Table: Effect of DAs on Greg (TN)**
> | Method | Webs       | TMN        | DBpedia    | R8         |
> |-----------|------------|------------|------------|------------|
> | Random to Similar | 12.5 ± 1.0 | 9.1 ± 0.7 | 15.6 ± 0.6 | 15.3 ± 1.0 |
> | Highest to Similar | **12.7 ± 0.8** | **9.6 ± 0.8** | **15.7 ± 0.5** | **15.5 ± 0.7** |
> | Lowest to Similar | 10.8 ± 0.3 | 7.4 ± 0.5 | 13.8 ± 0.5 | 14.5 ± 1.3 |
> | Both to Similar | 11.9 ± 0.9 | 8.4 ± 0.9 | 15.0 ± 0.9 | 14.9 ± 0.9 |
>
> Based on the results, “Highest to Similar” obtains the highest quality in most of the target domains, which shows the best generalization capability among the DAs when applying to Greg.
>
> &nbsp;
>
> For other requested changes:
> 1. When computing the distance between document representations (i.e., topic distributions), TopicalOT incorporates the semantic meaning of the support of the distribution—i.e, the topics—through the use of a cost matrix. This matrix is constructed based on the pairwise Optimal Transport (OT) distances between topics, thereby capturing their semantic relationships. In contrast, conventional distance metrics such as Euclidean, Cosine, and Hellinger distances disregard this semantic structure, treating topics as independent dimensions without accounting for their inherent meaning.
>
> 2. Thank you for pointing out the issues related to the figures, writing, and reference formatting. We will address these and make the necessary corrections in the revised version.

---

> > ### Comment · Reviewer_JJqT · 2025-04-28
> >
> > Dear authors, thank you for addressing my previous comments. With respect to Weakness 3, I meant, why can the word-level document augmentation approaches presented in Table 1 be used to produce a random augmentation that is **semantically similar** to the original document? For instance, the approach of “Random Drop” randomly samples $n$ words from the document and drops them. When some keywords related to the semantics of the original document are dropped, the new document is largely no longer semantically similar. The approaches such as “Random Insertion” and “Random to Dissimilar” face the same problem.

---

> ### Author Response · Authors · 2025-04-28
> **reply**
>
> That is correct, and we agree with your point. The word-level document augmentations listed in Table 1 are the approaches we focus on and explore (see Appendix D.3). In our implementation, we apply only the "highest to similar" strategy within Greg. We will revise our wording to clarify this in the paper.
>
> Thank you for pointing out this issue!

---

### Review · Reviewer_1C2k · 2025-04-13

**Summary Of Contributions:**

This paper enhanced neural topic models (NTMs) in how well NTMs trained on one corpus can generalize to unseen corpora. The authors propose a model-agnostic framework that enhanced NTMs to encourage similar documents to have similar topic distributions. This is achieved by a regularization technique that minimizes the topical optimal transport distance between each original document and its augmented counterpart.

**Audience:**

No

**Claims And Evidence:**

No

**Requested Changes:**

See Strengths And Weaknesses

**Strengths And Weaknesses:**

Strengths

- The paper tackles the important problem of improving the generalization ability of Neural Topic Models (NTMs)
- The proposed method leads to clear performance improvements in cross-corpus settings, demonstrating its effectiveness.

Weaknesses
- NTMs as a research direction have become somewhat outdated, and from a practical standpoint, it’s difficult to see strong real-world value. For instance, it would be helpful to evaluate how well modern large language models like GPT-4o can perform on the same tasks addressed in the paper, and what accuracy they can achieve. Notably, the most recent citation in the introduction dates back to 2023, suggesting that the field has seen little recent progress. This raises concerns about the practical relevance of the work.
- In Section 5.1.3, all backbones are based on models from 2021 or earlier. Incorporating experiments with more modern architectures, such as LLaMA, would significantly strengthen the empirical validation.
- Finally, it remains unclear how much computational overhead the proposed method introduces. A quantitative analysis of its cost would be helpful.

---

> ### Author Response · Authors · 2025-04-24
> **Response to Reviewer 1C2k**
>
> We sincerely thank you for your comments and constructive suggestions.
>
> For Weaknesses 1 and 2, please refer to our global response (first section).
>
> For Weakness 3, please refer to our global response (second section).

---

### Author Response · Authors · 2025-04-24
**Global response to reviewers**

We sincerely appreciate all the reviewers' insightful comments and constructive suggestions—particularly the common recommendation to include additional and more recent baselines in our experiments, as well as to provide an analysis of the computational cost associated with applying Greg.

&nbsp;

In response to the request for additional baselines, we have conducted further experiments incorporating several topic models suggested by the reviewers, including:
* Traditional non-neural topic models such as LDA [1] and NMF [2];
* A recent neural topic model (NTM), ECRTM [3];
* A popular clustering-based topic model, BERTopic [4];
* Latest state-of-the-art LLM-based topic models such as TopicGPT [5] and LLM-ITL [6].

Additionally, we apply Greg to ECRTM and LLM-ITL, both of which incorporate an NTM component. We present the additional results as follows:

**Table: One source (20News) to different targets (CA)**
| Model     | Webs       | TMN        | DBpedia    | R8         |
|-----------|------------|------------|------------|------------|
| LDA       | 47.7 ± 1.0 | 48.2 ± 0.7 | 38.4 ± 1.0 | 69.2 ± 0.9 |
| NMF       | 52.4 ± 0.6 | 47.7 ± 0.4 | 47.1 ± 0.8 | 76.9 ± 0.4 |
| BERTopic  | 42.5 ± 1.7 | 39.0 ± 0.4 | 30.0 ± 1.5 | 71.8 ± 3.2 |
| TopicGPT  | 42.8 ± 0.0 |	55.7 ± 0.0 | 38.9 ± 0.0 | 55.4 ± 0.0 |
|ECRTM| 58.4 ± 0.8 | 53.2 ± 0.5 | 53.0 ± 1.3 | 76.4 ± 1.4 |
| ECRTM + Greg | 64.0 ± 0.4 | 60.2 ± 0.6 | 58.7 ± 1.2 | 78.8 ± 2.4|
| LLM-ITL | 58.2 ± 0.6 | 53.4 ± 0.6 | 52.6 ± 0.7 | 79.0 ± 1.1 |
| LLM-ITL + Greg | 65.2 ± 0.5 | 60.6 ± 0.8 | 61.2 ± 2.2 | 79.1 ± 1.7|

&nbsp;

**Table: One source (20News) to different targets (TP)**
| Model     | Webs       | TMN        | DBpedia    | R8         |
|-----------|------------|------------|------------|------------|
| LDA       | 48.1 ± 1.1 | 45.9 ± 0.9 | 30.8 ± 1.6 | 60.4 ± 2.4|
| NMF       | 44.3 ± 1.5 | 41.1 ± 0.3 | 31.1 ± 0.5 | 67.5 ± 0.7|
| BERTopic  | 38.4 ± 1.4 | 38.4 ± 0.6 | 25.5 ± 0.6 | 61.7 ± 2.5|
| TopicGPT  | 44.6 ± 0.0 | 56.0 ± 0.0 | 39.5 ± 0.0 | 55.8 ± 0.0|
|ECRTM| 40.8 ± 1.8 | 40.7 ± 1.6 | 21.3 ± 1.1 | 57.0 ± 0.9|
| ECRTM + Greg | 45.9 ± 3.6 | 49.5 ± 2.0 | 22.6 ± 1.2 | 58.4 ± 0.3 |
| LLM-ITL | 40.8 ± 3.9 | 41.9 ± 1.3 | 22.6 ± 1.0 | 60.3 ± 1.3 |
| LLM-ITL + Greg | 44.9 ± 1.4 | 48.6 ± 1.1 | 22.9 ± 1.8 | 58.8 ± 3.2 |


&nbsp;

**Table: One source (20News) to different targets (TN)**
| Model     | Webs       | TMN        | DBpedia    | R8         |
|-----------|------------|------------|------------|------------|
| LDA       | 20.1 ± 0.6 | 14.7 ± 0.5 | 19.5 ± 0.8 | 16.1 ± 0.8 |
| NMF       | 17.1 ± 0.4 | 10.9 ± 0.1 | 20.7 ± 0.2 | 20.3 ± 1.4 |
| BERTopic  | 15.1 ± 0.8 | 9.2 ± 0.5 | 16.5 ± 0.5 | 17.3 ± 3.0 |
| TopicGPT  | 26.6 ± 0.0 | 33.4 ± 0.0 | 37.1 ± 0.0 | 12.2 ± 0.0 |
|ECRTM| 18.4 ± 0.7 | 13.2 ± 0.9 | 14.9 ± 0.2 | 12.0 ± 1.1 |
| ECRTM + Greg | 23.6 ± 1.7 | 19.5 ± 0.9 | 16.6 ± 1.3 | 13.1 ± 1.5 |
| LLM-ITL | 18.8 ± 1.5 | 13.9 ± 0.7 | 15.7 ± 0.9 | 15.5 ± 1.3 |
| LLM-ITL + Greg | 23.4 ± 0.8 | 19.2 ± 0.6 | 16.8 ± 1.2 | 13.8 ± 1.9 |

From the updated results and our previous findings, we observe the following: 1. Applying Greg to NTMs consistently improves document representations quality on various unseen target datasets across multiple evaluation metrics. 2. These benefits also extend to recent models such as ECRTM and LLM-ITL. 3. The enhanced document representation quality on unseen target datasets as measured by CA, surpasses that of LLM-based topic models such as TopicGPT and LLM-ITL in most cases.

Moreover, we are in the process of completing the experiments for these additional baselines on the remaining experimental settings, and the updated results and analysis will be included in the revised version of the paper.

&nbsp;

Regarding the computational cost of applying Greg, we conducted an additional quantitative analysis. Specifically, we recorded the running time of different models and computed the average running time per epoch. All experiments were conducted on the same machine equipped with a single RTX 6000 GPU and an Intel i9-14900K CPU. The results are illustrated as follows:

**Table: Running time of different models**
| Model     | Avg. Per Epoch (mm:ss)|
|-----------|------------|
| NVDM| 00:01 |
NVDM + Greg | 01:27 |
PLDA + Greg | 01:17 |
SCHOLAR + Greg | 01:03 |
CLNTM + Greg | 01:02 |
ECRTM + Greg | 01:51 |
LLM-ITL | 03:06 |

From the results, we observe that applying Greg incurs a computational cost of less than 2 minutes per epoch in our experiments. We believe this running time is generally acceptable in practice—particularly in the era of large language models (LLMs). For example, the LLM-based model (LLM-ITL) in our experiments requires around 3 minutes per epoch, which is more computationally expensive than Greg; TopicGPT relies on OpenAI's GPT models, which incur usage costs—making it particularly expensive when applied to large-scale datasets.

---

> ### Author Response · Authors · 2025-04-24
> **Reference list**
>
> [1] Blei, D. M., Ng, A. Y., & Jordan, M. I. (2003). Latent dirichlet allocation. Journal of machine Learning research, 3(Jan), 993-1022.
>
> [2] Févotte, C., & Idier, J. (2011). Algorithms for nonnegative matrix factorization with the β-divergence. Neural computation, 23(9), 2421-2456.
>
> [3] Wu, X., Dong, X., Nguyen, T. T., & Luu, A. T. (2023, July). Effective neural topic modeling with embedding clustering regularization. In International Conference on Machine Learning (pp. 37335-37357). PMLR.
>
> [4] Grootendorst, M. (2022). BERTopic: Neural topic modeling with a class-based TF-IDF procedure. arXiv preprint arXiv:2203.05794.
>
> [5] Pham, C. M., Hoyle, A., Sun, S., Resnik, P., & Iyyer, M. (2023). Topicgpt: A prompt-based topic modeling framework. arXiv preprint arXiv:2311.01449.
>
> [6] Yang, X., Zhao, H., Xu, W., Qi, Y., Lu, J., Phung, D., & Du, L. (2024). Neural Topic Modeling with Large Language Models in the Loop. arXiv preprint arXiv:2411.08534.

---

### Decision · Action_Editor_9Cds · 2025-06-18

**Recommendation:** Reject

**Additional Comments:**

1. Expand evaluation with extra topic quality metrics and analyze result discrepancies.
2. Detail computational cost analysis and benchmark against other models.
3. Emphasize difference relative to recent work and conduct an exhaustive recent literature review.

For more details, please refer to the reviews to improve the manuscript.

**Audience:**

Yes

**Audience Explanation:**

The topic of improving the generalization ability of neural topic models across corpora is relevant to the machine learning community. However, the current execution and presentation of the work need significant improvements.

**Claims And Evidence:**

No

**Claims Explanation:**

The reviewers pointed out several issues. The evaluation metrics used may not fully capture the generalization ability of neural topic models across corpora. The topic quality metric NPMI showed inconsistent results compared to recent literature. The computational cost analysis was also not detailed enough. More rigorous evaluation and comprehensive analysis are needed.

**Resubmission Of Major Revision:**

The authors may consider submitting a major revision at a later time.